# FARE: PROVABLY FAIR REPRESENTATION LEARNING

## ABSTRACT

Fair representation learning (FRL) is a popular class of methods aiming to produce fair classifiers via data preprocessing. However, recent work has shown that prior methods achieve worse accuracy-fairness tradeoffs than originally suggested by their results. This dictates the need for FRL methods that provide provable upper bounds on unfairness of any downstream classifier, a challenge yet unsolved. In this work we address this challenge and propose Fairness with Restricted Encoders (FARE), the first FRL method with provable fairness guarantees. Our key insight is that restricting the representation space of the encoder enables us to derive suitable fairness guarantees, while allowing empirical accuracy-fairness tradeoffs comparable to prior work. FARE instantiates this idea with a tree-based encoder, a choice motivated by inherent advantages of decision trees when applied in our setting. Crucially, we develop and apply a practical statistical procedure that computes a high-confidence upper bound on the unfairness of any downstream classifier. In our experimental evaluation on several datasets and settings we demonstrate that FARE produces tight upper bounds, often comparable with empirical results of prior methods, which establishes the practical value of our approach.

## 1 INTRODUCTION

It has been repeatedly shown that machine learning systems deployed in real-world applications propagate training data biases, producing discriminatory predictions (Buolamwini & Gebru, 2018; Corbett-Davies et al., 2017; Kleinberg et al., 2017; Tatman & Kasten, 2017). This is especially concerning in decision-making applications on data that represents humans (e.g., financial or medical), and can lead to unfavorable treatment that negatively affects certain subgroups of the population (Brennan et al., 2009; Khandani et al., 2010; Barocas & Selbst, 2016). For instance, a loan prediction system deployed by a financial institution might recommend loan rejection based on a *sensitive attribute* of a client, such as race or gender. These observations have forced regulators into action, leading to directives (FTC, 2021; EU, 2021) which demand parties aiming to deploy such systems to ensure *fairness* (Dwork et al., 2012) of their predictions. Mitigation of unfairness has become a key concern for organizations, with the highest increase in perceived relevance over the previous year, out of all potential risks of artificial intelligence (Chui et al., 2021; Benaich & Hogarth, 2021).

**Fair representation learning** A promising approach that attempts to address this issue is *fair representation learning* (FRL) (Zemel et al., 2013; Moyer et al., 2018; Madras et al., 2018; Gupta et al., 2021; Kim et al., 2022; Shui et al., 2022; Balunović et al., 2022)—a long line of work that preprocesses the data using an encoder $f$, transforming each datapoint $x \in \mathcal{X}$ into a debiased representation $z$. The key promise of FRL is that these debiased representations can be given to other parties, who want to solve a prediction task without being aware of fairness (or potentially even being fine with discriminating), while ensuring that *any* downstream classifier they train on these representations has favorable fairness. However, recent work (Xu et al., 2020; Song & Shmatikov, 2020; Gupta et al., 2021) has demonstrated that for some FRL methods it is possible to train significantly more unfair classifiers than originally claimed. This illuminates a major drawback of all existing work—their claim about fairness of the downstream classifiers holds only for the models they considered during the evaluation, and does not *guarantee* favorable fairness of other downstream classifiers trained on $z$. This is insufficient for critical applications where fairness must be guaranteed or is enforced by regulations, leading to our key question:

*Can we create an FRL method that provably bounds the unfairness of any downstream classifier?*

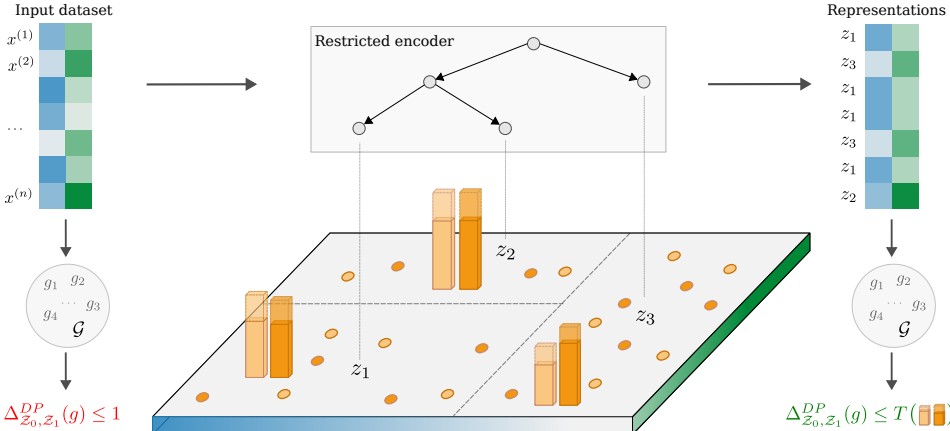

Figure 1: Overview of our provably fair representation learning method, FARE. The input dataset is transformed into fair representations using a restricted encoder. Our method can compute a provable upper bound $T$ on unfairness of any classifier $g \in \mathcal{G}$ trained on these representations.

The most prominent prior attempt to tackle this question, and the work most closely related to ours, is FNF (Balunović et al., 2022); we discuss other related work in Section 2. Assuming two groups $s = 0$ and $s = 1$ based on the sensitive attribute $s$, FNF shows that knowing the input distribution for each group can lead to an upper bound on unfairness of any downstream classifier. While this work is an important step towards provable fairness, the required assumption is unrealistic for most machine learning settings, and represents an obstacle to applying the approach in practice. Thus, the original problem of creating FRL methods that provide fairness guarantees remains largely unsolved.

**This work: provably fair representation learning** We propose FARE (Fairness with Restricted Encoders, Fig. 1)—the first FRL method that offers provable upper bounds on the unfairness of any downstream classifier $g$ trained on its representations, without unrealistic prior assumptions. Our key insight is that using an encoder with *restricted representations*, i.e., limiting possible representations to a finite set $\{z_1, \ldots, z_k\}$, allows us to derive a practical statistical procedure that computes a high-confidence upper bound on the unfairness of any $g$, detailed in Section 4. FARE instantiates this idea with a suitable encoder based on fair decision trees (see Section 5), leading to a practical end-to-end FRL method which produces debiased representations augmented with strong fairness guarantees.

More concretely, FARE takes as input the set of samples $\{x^{(1)}, \ldots, x^{(n)}\}$ from the input distribution $\mathcal{X}$ (left of Fig. 1), and partitions the input space into $k$ *cells* (middle plane, $k = 3$ in this example) using the decision tree encoder. Finally, all samples from the same cell $i$ are transformed into the same representation $z_i$ (right). As usual in FRL, training a downstream classifier on representations leads to lower empirical unfairness, while slightly sacrificing accuracy on the prediction task.

However, the main advantage of FARE comes from the fact that using a restricted set of representations allows us to, using the given samples, estimate the distribution of two sensitive groups in each cell, i.e., compute an empirical estimate of the conditional probabilities $P(s = 0|z_i)$ and $P(s = 1|z_i)$ (solid color orange bars) for all $z_i$. Further, we can use confidence intervals to obtain upper bounds on these values that hold with high probability (transparent bars). As noted above, this in turn leads to the key feature of our method: a tight upper bound $T$ on the unfairness of any $g \in \mathcal{G}$, where $\mathcal{G}$ is the set of all downstream classifiers that can be trained on the resulting representations. As we later elaborate on, increasing the number of samples $n$ makes the bounds tighter. Given the current trend of rapidly growing datasets, this further illustrates the practical value of FARE.

In our experimental evaluation in Section 6 we empirically demonstrate that on real datasets FARE produces tight upper bounds, i.e., the unfairness of any downstream classifier trained on FARE representations is tightly upper-bounded, which was not possible for any of the previously proposed FRL methods. Moreover, these downstream classifiers are able to achieve comparable empirical accuracy-fairness tradeoffs to methods from prior work. We believe this work represents a major step towards solving the important problem of preventing discriminatory machine learning models.

**Main contributions**    The key contributions of our work are:

- A practical statistical procedure that, for restricted encoders, upper-bounds the unfairness of any downstream classifier trained on their representations (Section 4).

- An end-to-end FRL method FARE, that instantiates this approach with a fair decision tree encoder (Section 5), and applies the said statistical procedure to augment the representations with a tight provable upper bound on unfairness of any downstream classifier.

- An extensive experimental evaluation in several settings, demonstrating favorable empirical fairness results, as well as tight upper bounds on unfairness (which were out of reach for prior work), often comparable to empirical results of existing FRL methods (Section 6).

## 2    RELATED WORK

We discuss related work on FRL, prior attempts to obtain fairness guarantees, and fair decision trees.

**FRL for group fairness**    Following Zemel et al. (2013) which originally introduced FRL, a plethora of different methods have been proposed based on optimization (Calmon et al., 2017; Shui et al., 2022), adversarial training (Edwards & Storkey, 2016; Xie et al., 2017; Madras et al., 2018; Song et al., 2019; Feng et al., 2019; Roy & Boddeti, 2019; Jaiswal et al., 2020; Kairouz et al., 2022; Kim et al., 2022), variational approaches (Louizos et al., 2016; Moyer et al., 2018; Oh et al., 2022; Liu et al., 2022), disentanglement (Sarhan et al., 2020), mutual information (Gupta et al., 2021; Gitiaux & Rangwala, 2022), and normalizing flows (Balunović et al., 2022; Cerrato et al., 2022). Notably, no prior method restricts representations as defined in Section 4, which is a key step in our work. While Zemel et al. (2013) map data to a set of *prototypes*, this mapping is probabilistic, and thus fundamentally incompatible with our bounding procedure (see Section 4 for further details).

**Towards fairness guarantees**    The key issue is that most of these methods produce representations that have no provable guarantees of fairness, meaning a model trained on their representations could have arbitrarily bad fairness. In fact, prior work (Elazar & Goldberg, 2018; Xu et al., 2020; Gupta et al., 2021) has shown that methods based on adversarial training often significantly overestimate the fairness of their representations. Some of these works (McNamara et al., 2017; Zhao et al., 2020; Gupta et al., 2021; Shen et al., 2021; Kairouz et al., 2022) propose theoretically-principled FRL, but do not provide provable fairness guarantees with finite samples (we discuss the difference in Appendix E). Closest to our work is FNF (Balunović et al., 2022) that can compute high-confidence bounds, but critically, assumes knowledge of the input probability distribution, which is rarely the case in practice. Our work makes no such assumption, which makes it significantly more practical.

**Provable fairness in other settings**    Numerous related works on provable fairness provide a different kind of guarantee or assume a different setting than ours. First, several FRL methods have proposed approaches for learning individually fair representations (Lahoti et al., 2019; Ruoss et al., 2020; Peychev et al., 2021), a different notion of fairness than group fairness which we focus on. Prior work has also examined provable fairness guarantees in various problem settings such as ranking (Konstantinov & Lampert, 2021), distribution shifting (Kang et al., 2022; Jin et al., 2022), fair classification with in-processing (Feldman et al., 2015; Donini et al., 2018; Celis et al., 2019), individually fair classification with post-processing (Petersen et al., 2021), and fair meta-learning (Oneto et al., 2020). These are all different from our setting, which is FRL for group fairness.

**Fair decision trees**    The line of work focusing on adapting decision trees to fairness concerns includes a wide range of methods which differ mainly in the branching criterion. Common choices include variations of Gini impurity (Kamiran et al., 2010; Raff et al., 2018; Zhang & Ntoutsi, 2019), mixed-integer programming (Aghaei et al., 2019; Wang et al., 2022) or AUC (Barata & Veenman, 2021), while some apply adversarial training (Grari et al., 2020; Ranzato et al., 2021). Further, some works operate in a different setting such as online learning (Zhang & Ntoutsi, 2019) or post-processing (Abebe et al., 2022). The only works in this area that offer provable fairness guarantees are Ranzato et al. (2021), which certifies individual fairness for post-processing, and Meyer et al. (2021), which certifies that predictions will not be affected by data changes. This fundamentally differs from our FRL setting where the goal is to certify fairness of any downstream classifier.

## 3 PRELIMINARIES

We now set up the notation and provide the background necessary to understand our contributions.

**Fair representation learning**    Assume data $(\boldsymbol{x}, s) \in \mathbb{R}^d \times \{0, 1\}$ from a joint probability distribution $\mathcal{X}$, where each datapoint belongs to a group with respect to a sensitive attribute $s$. While in this work we focus on the case of binary $s$, our results can be directly extended to other settings. Further, we focus on binary classification, i.e., given a label $y \in \{0, 1\}$ associated with each datapoint, the goal is to build a model $g \colon \mathbb{R}^d \to \{0, 1\}$ that predicts $y$ from $\boldsymbol{x}$. Besides maximizing accuracy of $g$, we aim to maximize its fairness with respect to sensitive groups, according to some fairness definition, which often implies a slight loss in accuracy as these two goals are generally at odds.

A large class of methods aims to directly produce $g$ with satisfactory fairness properties. A different group of methods, our focus here, preprocesses data by applying an encoder $f \colon \mathbb{R}^d \to \mathbb{R}^{d'}$ to obtain a new *representation* $\boldsymbol{z} = f(\boldsymbol{x}, s)$ of each datapoint. This induces a joint distribution $\mathcal{Z}$ of $(\boldsymbol{z}, s)$. The downstream classifier $g$ is now trained to predict $y$ from $\boldsymbol{z}$, i.e., now we have $g \colon \mathbb{R}^{d'} \to \{0, 1\}$. The main advantage of these methods is that by ensuring fairness properties of representations $\boldsymbol{z}$, we can limit the unfairness of *any* $g$ trained on data from $\mathcal{Z}$.

**Fairness metric**    Let $\mathcal{Z}_0$ and $\mathcal{Z}_1$ denote conditional distributions of $\boldsymbol{z}$ where $s = 0$ and $s = 1$, respectively. In this work, we aim to minimize the *demographic parity distance* of $g$, reflecting the goal of equally likely assigning positive outcomes to inputs from both sensitive groups:

$$\Delta_{\mathcal{Z}_0, \mathcal{Z}_1}^{DP}(g) := |\mathbb{E}_{\boldsymbol{z} \sim \mathcal{Z}_0}[g(\boldsymbol{z})] - \mathbb{E}_{\boldsymbol{z} \sim \mathcal{Z}_1}[g(\boldsymbol{z})]| \,.$$

Our choice of metric is primarily motivated by consistency with prior work—other definitions (e.g., equalized odds) may be more suitable for a particular use-case (Hardt et al., 2016), and our method can be easily adapted to support them, following the corresponding results of Madras et al. (2018).

In the remainder of this work, we will use $p_0$ and $p_1$ to denote the PDFs of $\mathcal{Z}_0$ and $\mathcal{Z}_1$ respectively, i.e., $p_0(\boldsymbol{z}_i) = P(\boldsymbol{z}_i | s = 0)$ and $p_1(\boldsymbol{z}_i) = P(\boldsymbol{z}_i | s = 1)$ and $p$ to denote the PDF of the marginal distribution of $\boldsymbol{z}$. Similarly, we will use $q$ for the marginal distribution of $s$, and $q_i$ for the conditional distribution of $s$ for $\boldsymbol{z} = z_i$, i.e., $q_i(0) = P(s = 0 | \boldsymbol{z} = z_i)$ and $q_i(1) = P(s = 1 | \boldsymbol{z} = z_i)$.

**Classification trees**    Starting from the training set $D_{root}$ of examples $(\boldsymbol{x}, y) \in \mathbb{R}^d \times \{0, 1\}$, a binary classification tree $f$ repeatedly *splits* some leaf node $P$ with assigned $D_P$, i.e., picks a split feature $j \in \{1, \ldots, d\}$ and a split threshold $v$, and adds two nodes $L$ and $R$ as children of $P$, such that $D_L = \{(\boldsymbol{x}, y) \in D_P \mid x_j \leq v\}$ and $D_R = D_P \setminus D_L$. $j$ and $v$ are picked to minimize a chosen criterion, weighted by $|D_L|$ and $|D_R|$, aiming to produce $y$-homogeneous leaves. We focus on Gini impurity, computed as $Gini_y(D) = 2p_y(1 - p_y) \in [0, 0.5]$ where $p_y = \sum_{(\boldsymbol{x}, y) \in D} \mathbb{1}\{y = 1\}/|D|$. At inference, a test example $\boldsymbol{x}$ is propagated to a leaf $l$, and we predict the majority class of $D_l$.

## 4 PROVABLE FAIRNESS BOUNDS FOR RESTRICTED ENCODERS

In the following we describe our primary contribution, the derivation of provable fairness bounds on unfairness of downstream classifiers, under the assumption of restricted encoders (i.e., encoders with restricted representations, explained in more detail shortly). In Section 5 we demonstrate the feasibility of our approach, by instantiating it with a particular encoder based on decision trees.

**Optimal adversary**    Consider the adversary $h \colon \mathbb{R}^{d'} \to \{0, 1\}$ predicting group membership $s$, which aims to maximize the following balanced accuracy objective:

$$BA_{\mathcal{Z}_0, \mathcal{Z}_1}(h) := \frac{1}{2} \left( \mathbb{E}_{\boldsymbol{z} \sim \mathcal{Z}_0}[1 - h(\boldsymbol{z})] + \mathbb{E}_{\boldsymbol{z} \sim \mathcal{Z}_1}[h(\boldsymbol{z})] \right). \tag{1}$$

Let $h^\star$, such that for all $h$, $BA_{\mathcal{Z}_0, \mathcal{Z}_1}(h^\star) \geq BA_{\mathcal{Z}_0, \mathcal{Z}_1}(h)$, denote the *optimal adversary*. Intuitively, the optimal adversary predicts the group $s$ for which the likelihood of $\boldsymbol{z}$ under the corresponding distribution ($\mathcal{Z}_0$ or $\mathcal{Z}_1$) is larger. More formally, $h^\star(\boldsymbol{z}) = \mathbb{1}\{p_1(\boldsymbol{z}) \geq p_0(\boldsymbol{z})\}$, where $\mathbb{1}\{\phi\} = 1$ if $\phi$ holds, and 0 otherwise (see Balunović et al. (2022) for a proof). As shown in Madras et al. (2018),

$$\Delta_{\mathcal{Z}_0, \mathcal{Z}_1}^{DP}(g) \leq 2 \cdot BA_{\mathcal{Z}_0, \mathcal{Z}_1}(h^\star) - 1 \tag{2}$$

holds for any $g$, i.e., we can upper-bound the unfairness of any downstream classifier trained on data from $\mathcal{Z}$ by computing the balanced accuracy of the optimal adversary $h^\star$. Shen et al. (2021) also discuss connection between the balanced accuracy and total variation.

**Restricted encoders** Prior work is unable to utilize Eq. (2) to obtain a fairness guarantee, as using unconstrained neural network encoders generally makes it intractable to compute the densities $p_0(\boldsymbol{z})$ and $p_1(\boldsymbol{z})$ that define the optimal adversary $h^*$. Notably, Balunović et al. (2022) use normalizing flows, allowing computation of $p_0(\boldsymbol{z})$ and $p_1(\boldsymbol{z})$ under the assumption of knowing corresponding densities in the original distribution $\mathcal{X}$. In contrast, we propose a class of encoders for which we can derive a procedure that can upper-bound the RHS of Eq. (2), and thus the unfairness of $g$, without imposing any assumption in terms of knowledge of $\mathcal{X}$. We rely only on a set of samples $(\boldsymbol{z}, s) \sim \mathcal{Z}$, obtained by applying $f$ to samples $(\boldsymbol{x}, s) \sim \mathcal{X}$, which are readily available in the form of a given dataset.

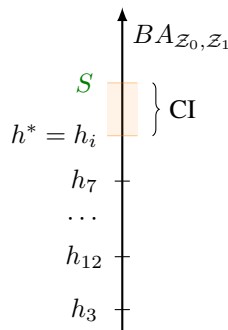

Figure 2: Restricted representations enable upperbounding of $BA_{\mathcal{Z}_0, \mathcal{Z}_1}(h^\star)$.

Namely, we hypothesize that restricting the space of representations can still lead to favorable fairness-accuracy tradeoffs. Based on this, we propose *restricted encoders* $f\colon \mathbb{R}^d \to \{\boldsymbol{z}_1, \ldots, \boldsymbol{z}_k\}$, i.e., encoders that map each $\boldsymbol{x}$ to one of $k$ possible values (*cells*) $\boldsymbol{z}_i \in \mathbb{R}^{d'}$. As now there is a finite number of possible values for a representation, we can use samples from $\mathcal{Z}$ to analyze the optimal adversary $h^*$ on each possible $\boldsymbol{z}$. Moreover, we can upper-bound its balanced accuracy on the whole distribution $\mathcal{Z}$ with some value $S$ with high probability, using confidence intervals (CI) (as illustrated in Fig. 2). Finally, we can apply Eq. (2) to obtain the bound $\Delta^{DP}_{\mathcal{Z}_0, \mathcal{Z}_1}(g) \leq 2S - 1 = T$.

A detailed presentation of our upper-bounding procedure follows. In Sections 5 and 6 we demonstrate how to design practical and efficient restricted encoders, by describing and evaluating our instantiation based on decision trees, which, as we later discuss, are naturally suited to this problem.

**Upper-bounding the balanced accuracy** Starting from Eq. (1), we reformulate the balanced accuracy of the optimal adversary as follows:

$$BA_{\mathcal{Z}_0, \mathcal{Z}_1}(h^*) = \frac{1}{2}\left(\sum_{i=1}^k p_0(\boldsymbol{z}_i) \cdot [1 - h^*(\boldsymbol{z}_i)] + \sum_{i=1}^k p_1(\boldsymbol{z}_i) \cdot [h^*(\boldsymbol{z}_i)]\right) \qquad (\mathbb{E} \text{ of discrete RV})$$

$$= \frac{1}{2}\left(\sum_{i=1}^k \max\left(p_0(\boldsymbol{z}_i), p_1(\boldsymbol{z}_i)\right)\right) \qquad (\text{Optimal adversary})$$

$$= \sum_{i=1}^k p(\boldsymbol{z}_i) \cdot \max\left(\underbrace{(1/2q(0))}_{\alpha_0} \cdot q_i(0), \underbrace{(1/2q(1))}_{\alpha_1} \cdot q_i(1)\right), \qquad (\text{Bayes' rule})$$

where applications of Bayes' rule are $p_0(\boldsymbol{z}_i) = q_i(0)p(\boldsymbol{z}_i)/q(0)$ and $p_1(\boldsymbol{z}_i) = q_i(1)p(\boldsymbol{z}_i)/q(1)$. As we do not know $\mathcal{Z}$, but instead have access to $n$ samples $(\boldsymbol{z}^{(j)}, s^{(j)}) \sim \mathcal{Z}$, we will aim to upper-bound $BA_{\mathcal{Z}_0, \mathcal{Z}_1}(h^*)$ with high probability. In particular, we focus on the final expression above, the prior-weighted (i.e., weighted by $p(\boldsymbol{z}_i)$) per-cell balanced accuracy (i.e., $\max(\alpha_0 q_i(0), \alpha_1 q_i(1))$ for each cell $i$), where we define $\alpha_0 = 1/2q(0)$ and $\alpha_1 = 1/2q(1)$.

Next, we introduce 3 lemmas, and later combine them to obtain the desired upper bound. We use $B(p; v, w)$ to denote the $p$-th quantile of a beta distribution with parameters $v$ and $w$. Note that for Lemma 1 we do not use the values $\boldsymbol{z}^{(j)}$ in the proof, but still introduce them for consistency.

**Lemma 1** (Bounding base rates). *Given $n$ independent samples $\{(\boldsymbol{z}^{(1)}, s^{(1)}), \ldots, (\boldsymbol{z}^{(n)}, s^{(n)})\} \sim \mathcal{Z}$ and a parameter $\epsilon_b$, for $\alpha_0$ and $\alpha_1$ as defined above, it holds that*

$$\alpha_0 < \frac{1}{2B(\frac{\epsilon_b}{2}; m, n - m + 1)}, \quad \text{and} \quad \alpha_1 < \frac{1}{2(1 - B(1 - \frac{\epsilon_b}{2}; m + 1, n - m))},$$

*with confidence $1 - \epsilon_b$, where $m = \sum_{j=1}^n \mathbb{1}\{s^{(j)} = 0\}$.*

*Proof.* We define $n$ independent Bernoulli random variables $X^{(j)} := \mathbb{1}\{s^{(j)} = 0\}$ with same unknown success probability $q(0)$. Using the Clopper-Pearson binomial CI (Clopper & Pearson, 1934) (Appendix A) to estimate the probability $q(0)$ we get $P(q(0) \leq B(\frac{\epsilon_b}{2}; m, n - m + 1)) \leq \epsilon_b/2$ and $P(q(0) \geq B(1 - \frac{\epsilon_b}{2}; m + 1, n - m)) \leq \epsilon_b/2$. Substituting $q(0) = 1 - q(1)$ in the latter, as well as the definitions of $\alpha_0$ and $\alpha_1$ in both inequalities, produces the inequalities from the lemma statement, which per union bound simultaneously hold with confidence $1 - \epsilon_b$. $\square$

**Lemma 2** (Bounding balanced accuracy for each cell). *Given $n$ independent samples $\{(\boldsymbol{z}^{(1)}, s^{(1)}), \ldots, (\boldsymbol{z}^{(n)}, s^{(n)})\} \sim \mathcal{Z}$, parameter $\epsilon_c$, and constants $\bar{\alpha}_0$ and $\bar{\alpha}_1$ such that $\alpha_0 < \bar{\alpha}_0$ and $\alpha_1 < \bar{\alpha}_1$, it holds for each cell $i \in \{1, \ldots, k\}$, with total confidence $1 - \epsilon_c$, that*

$$\max(\alpha_0 \cdot q_i(0), \alpha_1 \cdot q_i(1)) \leq t_i, \tag{3}$$

*where $t_i = \max\left(\bar{\alpha}_0 B(\frac{\epsilon_c}{2k}; m_i, n_i - m_i + 1), \bar{\alpha}_1(1 - B(1 - \frac{\epsilon_c}{2k}; m_i + 1, n_i - m_i))\right)$. In this expression, $n_i = |Z_i|$, and $m_i = \sum_{j \in Z_i} \mathbb{1}\{s^{(j)} = 0\}$, where we denote $Z_i = |\{j | \boldsymbol{z}^{(j)} = \boldsymbol{z}_i\}|$.*

*Proof.* As in Lemma 1, for each cell we use the Clopper-Pearson CI to estimate $q_i(0)$ with samples indexed by $Z_i$ and confidence $1 - \epsilon_c/k$. As before, we apply $q_i(0) = 1 - q_i(1)$ to arrive at a set of $k$ inequalities of the form Eq. (3), which per union bound jointly hold with confidence $1 - \epsilon_c$. $\square$

**Lemma 3** (Bounding the sum). *Given $n$ independent samples $\{(\boldsymbol{z}^{(1)}, s^{(1)}), \ldots, (\boldsymbol{z}^{(n)}, s^{(n)})\} \sim \mathcal{Z}$, where for each $j \in \{1, \ldots, n\}$ we define a function $idx(\boldsymbol{z}^{(j)}) = i$ such that $\boldsymbol{z}^{(j)} = \boldsymbol{z}_i$ (cell index), parameter $\epsilon_s$, and a set of real-valued constants $\{t_1, \ldots, t_k\}$, it holds that*

$$P\left(\sum_{i=1}^{k} p(\boldsymbol{z}_i)t_i \leq S\right) \geq 1 - \epsilon_s, \text{where } S = \frac{1}{n}\sum_{j=1}^{n} t_{idx(\boldsymbol{z}^{(j)})} + (b - a)\sqrt{\frac{-\log \epsilon_s}{2n}}, \tag{4}$$

*and we denote $a = \min\{t_1, \ldots, t_k\}$ and $b = \max\{t_1, \ldots, t_k\}$.*

*Proof.* For each $j$ let $X^{(j)} := t_{idx(\boldsymbol{z}^{(j)})}$ denote a random variable. As for all $j$, $X^{(j)} \in [a, b]$ with probability 1 and $X^{(j)}$ are independent, we can apply Hoeffding's inequality (Hoeffding, 1963) (restated in Appendix A) to upper-bound the difference between the population mean $\sum_{i=1}^{k} p(\boldsymbol{z}_i)t_i = \mathbb{E}_{\boldsymbol{z} \sim \mathcal{Z}} t_{idx(\boldsymbol{z})}$ and its empirical estimate $\frac{1}{n}\sum_{j=1}^{n} X^{(j)}$. Setting the upper bound such that the error is below $\epsilon_s$ directly recovers $S$ and the statement of the lemma. $\square$

**Applying the lemmas** Finally, we describe how we apply the lemmas in practice to upper-bound $BA_{\mathcal{Z}_0, \mathcal{Z}_1}(h^\star)$, and in turn upper-bound $\Delta_{\mathcal{Z}_0, \mathcal{Z}_1}^{DP}(g)$ for any downstream classifier $g$ trained on representations learned by a restricted encoder. We assume a standard setting, where a set $D$ of datapoints $\{(\boldsymbol{x}^{(j)}, s^{(j)})\}$ from $\mathcal{X}$ is split into a training set $D_{train}$, used to train $f$, validation set $D_{val}$, held-out for the upper-bounding procedure (and not used in training of $f$ in any capacity), and a test set $D_{test}$, used to evaluate the empirical accuracy and fairness of downstream classifiers.

After training the encoder and applying it to produce representations $(\boldsymbol{z}^{(j)}, s^{(j)}) \sim \mathcal{Z}$ for all three data subsets, we aim to derive an upper bound on $\Delta_{\mathcal{Z}_0, \mathcal{Z}_1}^{DP}(g)$ for any $g$, that holds with confidence at least $1 - \epsilon$, where $\epsilon$ is the hyperparameter of the procedure (we use $\epsilon = 0.05$). To this end, we heuristically choose some decomposition $\epsilon = \epsilon_b + \epsilon_c + \epsilon_s$, and apply Lemma 1 on $D_{train}$ to obtain upper bounds $\alpha_0 < \bar{\alpha}_0$ and $\alpha_1 < \bar{\alpha}_1$ with error probability $\epsilon_b$. As mentioned above, using $D_{train}$ in this step is sound as estimated probabilities $q(0)$ and $q(1)$ are independent of the encoder $f$. Next, we use $\bar{\alpha}_0, \bar{\alpha}_1$ and $D_{val}$ in Lemma 2, to obtain upper bounds $t_1, \ldots, t_k$ on per-cell accuracy that jointly hold with error probability $\epsilon_c$. Finally, we upper-bound the sum $\sum_{i=1}^{k} p(\boldsymbol{z}_i)t_i \leq S$ with error probability $\epsilon_s$ using Lemma 3 on $D_{test}$ with previously computed $t_1, \ldots, t_k$. Combining this with Eq. (2) finally gives the desired upper bound

$$\Delta_{\mathcal{Z}_0, \mathcal{Z}_1}^{DP}(g) \leq 2 \cdot BA_{\mathcal{Z}_0, \mathcal{Z}_1}(h^\star) - 1 \leq 2S - 1 = T, \tag{5}$$

which per union bound holds with desired error probability $\epsilon$, with respect to the sampling process.

This completes the upper-bounding procedure, enabling provable fair representation learning with no restrictive assumptions, which addresses a major limitation of prior work. Our procedure can be applied to representations produced by any restricted encoder. In the following sections, we will describe a particular instantiation based on decision trees, and experimentally demonstrate that applying the above procedure produces tight unfairness upper bounds on real datasets.

## 5 RESTRICTED REPRESENTATIONS WITH FAIR DECISION TREES

Next, we describe a practical restricted encoder, enabling a favorable accuracy-fairness tradeoff (similar to prior work), while allowing application of our results from Section 4 to provably bound the fairness of downstream classifiers trained on the representations (unique to our work).

Our encoder is based on decision trees, a choice motivated by strong results of tree-based models on tabular data (Borisov et al., 2021), as well as their feature space splitting procedure, whose discrete behavior is inherently suitable for our requirement of restricted representations. In particular, we train a classification tree $f$ with $k$ leaves, and encode all examples that end up in leaf $i$ to the same representation $z_i$. We construct $z_i$ based on the set of training examples in leaf $i$, taking the median value for continuous, and the most common value for categorical features (thus in our case, $d' = d$).

**Fairness-aware criterion**   Common splitting criteria are aimed at maximizing accuracy by making the distribution of $y$ in each leaf highly unbalanced, e.g., $Gini_y(D) = 2p_y(1 - p_y) \in [0, 0.5]$ where $p_y = \sum_{(\boldsymbol{x}, y) \in D} \mathbb{1}\{y = 1\}/|D|$. Aiming to use such a tree as an encoder generally leads to high unfairness, making it necessary to introduce a direct way to prioritize more fair tree structures.

To this end, similar to Kamiran et al. (2010) and others (see discussion of related work in Section 2), we use the criterion $FairGini(D) = (1 - \gamma)Gini_y(D) + \gamma(0.5 - Gini_s(D)) \in [0, 0.5]$, where $Gini_s$ is defined analogously to $Gini_y$. The second term aims to *maximize $Gini_s(D)$*, i.e., make the distribution of $s$ in each leaf $i$ as close to uniform (making it challenging for the adversary to infer the value of $s$ based on $z_i$), while the hyperparameter $\gamma$ controls the accuracy-fairness tradeoff.

**Fairness-aware categorical splits**   While usual splits of the form $x_j \leq v$ (see Section 3) are suitable for continuous, they are inefficient for categorical (usually one-hot) variables, as only 1 category can be isolated. Consequently, this increases the number of cells and makes our fairness bounds loose. Instead, we represent $n_j$ categories for feature $j$ as integers $c \in \{1, 2, ..., n_j\}$. To avoid evaluating all $2^{n_j} - 1$ possible partitions, we sort the values by $p_y(c) = \sum_{(\boldsymbol{x}, y) \in D_c} \mathbb{1}\{y = 1\}/|D_c|$ where $D_c = \{\boldsymbol{x} \in D \mid \boldsymbol{x}_j = c\}$, and consider all prefix-suffix partitions (*Breiman shortcut*).

This ordering focuses on accuracy and is provably optimal for $FairGini(D)$ with $\gamma = 0$ (Breiman et al., 1984). However, as it ignores fairness, it is inefficient for $\gamma > 0$. To alleviate this, we generalize the Breiman shortcut, and explore all prefix-suffix partitions under several orderings. Namely, for several values of the parameter $q$, we split the set of categories $\{1, 2, \ldots, n_j\}$ in $q$-quantiles with respect to $p_s(c)$ (defined analogous to $p_y(c)$), and sort each quantile by $p_y(c)$ as before, interspersing $q$ obtained arrays to obtain the final ordering. Note that while this offers no optimality guarantees, it is an efficient way to consider both objectives, complementing our fairness-aware criterion.

We defer the discussion of the hyperparameters of our encoder to Appendix B.

## 6 EXPERIMENTAL EVALUATION

We evaluate our method, FARE, on several common fairness datasets, demonstrating that it produces representations with fairness-accuracy tradeoffs comparable to prior work, while for the first time offering provable fairness bounds. We further provide more insights into FARE, discuss the interpretability of the representations, and provide additional experiments on transfer learning.

**Experimental setup**   We consider common fairness datasets: Health (Kaggle, 2012) and two variants of ACSIncome (Ding et al., 2021), ACSIncome-CA (only California), and ACSIncome-US (US-wide, larger but more difficult due to distribution shift). The sensitive attributes are age and sex, respectively. We compare our method with the following recent FRL baselines (described in Section 2): LAFTR (Madras et al., 2018), CVIB (Moyer et al., 2018), FCRL (Gupta et al., 2021), FNF (Balunović et al., 2022), sIPM-LFR (Kim et al., 2022), and FairPath (Shui et al., 2022). We provide all omitted details regarding datasets, baselines, and our experimental setup, in Appendix B.

**Main experiments**   We explore the fairness-accuracy tradeoff of each method by running it with various hyperparameters. Each run produces representations, used to train a 1-hidden-layer neural

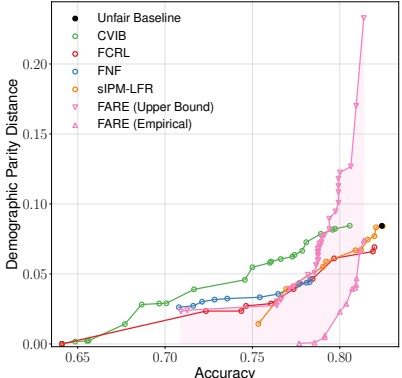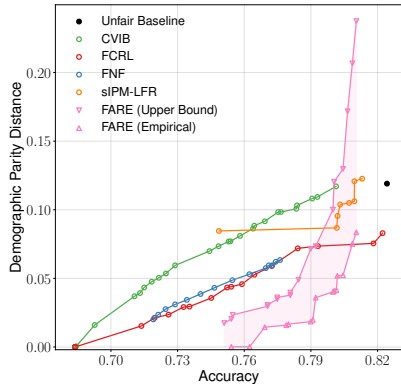

Figure 3: Evaluation of FRL methods on ACSIncome-CA (left) and ACSIncome-US (right).

network (1-NN) for the prediction task using a standard training procedure (same for each method), and plot its demographic parity (DP) distance and prediction accuracy.

Following Kim et al. (2022), we show a test set Pareto front for each method. Further, for FARE we independently show a Pareto front of a 95% confidence provable upper bound on DP distance (following Section 4), which is a key feature of our approach and cannot be produced by any other method. Finally, we include an Unfair Baseline, which uses an identity encoder. The results are shown in Figs. 3 and 4. We omit FairPath and LAFTR from the main plots (see extended results in Appendix C), as LAFTR has stability and convergence issues (Gupta et al., 2021; Kim et al., 2022), and FairPath uses a different metric to us (Shui et al., 2022).

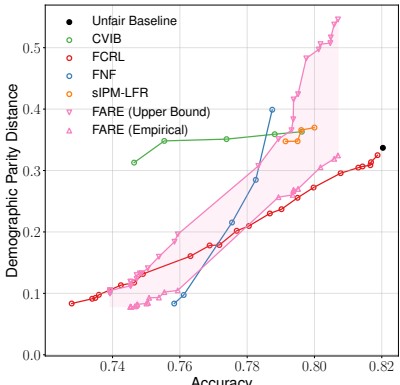

Figure 4: Evaluation on Health.

We observe that FARE can achieve a better (Fig. 3) or comparable (Fig. 4) accuracy-fairness tradeoff compared to baselines. Crucially, other methods cannot guarantee that there is no classifier with a worse DP distance when trained on their representations. This cannot happen for our method—we produce a *provable* upper bound on DP distance of *any* classifier trained on our representations. The results indicate that our provable upper bound is often comparable to *empirical* values of baselines. Note that there is a small gap (up to 1.5%) between the maximum accuracy of FARE and the unfair baseline, indicating a tradeoff of restricted encoders—FARE obtains provable bounds, but loses some information, limiting the predictive power of downstream classifiers. This is rarely an issue in practice, as obtaining meaningful fairness improvements generally requires a non-trivial accuracy loss, especially when $s$ is correlated with the task label. Finally, another advantage of FARE is its efficiency, with runtime of several seconds, as opposed to minutes or hours for baselines.

**Exploring downstream classifiers** In Fig. 5 (left), we show a representative point from Fig. 4, its fairness guarantee, and 24 diverse downstream classifiers (see Appendix B) trained on same representations, where half are trained to maximize accuracy, and half to maximize unfairness. The latter (left cluster) can reach higher unfairness than initially suggested, reaffirming a known limitation of prior work (Xu et al., 2020; Gupta et al., 2021): evaluating representations with some model class (here, a 1-NN) does not reliably estimate unfairness, as other classifiers (perhaps intentionally created by a malicious actor) might be more unfair. This highlights the value of FARE which provides a provable unfairness upper bound—all unfairness values still remain below a known upper bound.

Similarly, we explore a point from Fig. 3 (right), with accuracy 75.1% and DP distance of 0.005. As here $k = 6$, i.e., the possible representations are $\{z_1, \ldots, z_6\}$, the previous investigation of downstream classifiers simplifies. Instead of choosing a model class, we can enumerate all $2^6 = 64$ possible classifiers, and directly confirm that each DP distance is below the upper bound, as shown in Fig. 5 (middle). Note that in the original experiment, all baseline methods have DP distance $\geq 0.04$ at similar accuracy of $\approx 75\%$, implying that the FARE bound is in this case very tight.

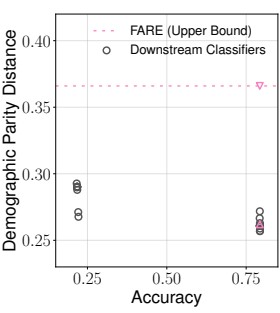 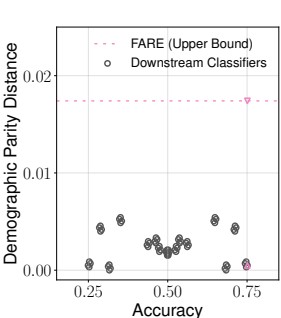 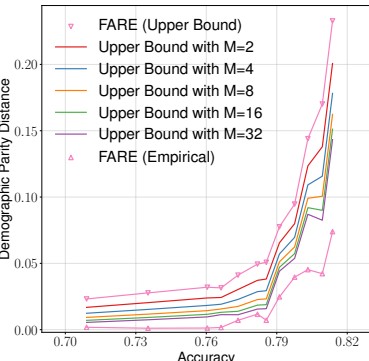

Figure 5: Comparing trained downstream classifiers with the FARE upper bound (left and middle). The impact of increasing the dataset size $M$ times on the fairness upper bound tightness (right).

**Data improves bounds**   In Fig. 3 we see that the FARE bounds are tighter for ACSIncome-US, as using more samples improves the bounding procedure. To investigate this further, we choose a representative set of FARE points from Fig. 3 (left), and repeat the upper-bounding procedure with the dataset repeated $M$ times, showing the resulting upper bounds for $M \in \{2, 4, 8, 16, 32\}$ in Fig. 5 (right). We can observe a significant improvement in the provable upper bound for larger dataset sizes, indicating that FARE is well-suited for large datasets and will benefit from ever-increasing amounts of data used in real-world machine learning deployments (Villalobos & Ho, 2022).

**Interpretability**   Another advantage of FARE is that its tree-based encoder enables direct interpretation of representations. To illustrate this, for representations with $k = 6$ analyzed in Fig. 5 (middle) we can easily find that, for example, the representation $z_6$ is assigned to each person older that $24$, with at least a Bachelor's degree, and an occupation in management, business or science.

**Transfer learning**   Finally, we analyze the transferability of learned representations across tasks. We produce a diverse set of representations on the Health dataset with each method, and following the procedure from prior work (Madras et al., 2018; Balunović et al., 2022; Kim et al., 2022) evaluate them on five unseen tasks $y$ (see Appendix B for details), where for each the goal is to predict a certain primary condition group. For each task and each method, we identify the highest accuracy obtained while keeping $\Delta_{\mathcal{Z}_0, \mathcal{Z}_1}^{DP}$ not above $0.20$ (or $0.05$). Moreover, we show $T$, the provable DP distance upper bound of FARE.

| $y$ | $\Delta_{\mathcal{Z}_0, \mathcal{Z}_1}^{DP}$ | $T$ | FARE | FCRL | FNF | sIPM |
|---|---|---|---|---|---|---|
| MIS | $\leq 0.20$ | 0.64 | 79.3 | 78.6 | 78.9 | 79.8 |
| | $\leq 0.05$ | 0.54 | 78.7 | 78.6 | 78.7 | 78.6 |
| NEU | $\leq 0.20$ | 0.64 | 73.2 | 72.4 | 71.9 | 76.6 |
| | $\leq 0.05$ | 0.42 | 72.1 | 71.4 | 71.7 | / |
| ART | $\leq 0.20$ | 0.41 | 74.4 | 70.7 | 68.9 | 78.3 |
| | $\leq 0.05$ | 0.23 | 69.5 | 69.5 | 68.5 | / |
| MET | $\leq 0.20$ | 0.46 | 69.8 | 69.2 | 75.0 | / |
| | $\leq 0.05$ | 0.12 | 66.1 | 65.3 | / | / |
| MSC | $\leq 0.20$ | 0.53 | 67.2 | 70.5 | 73.0 | / |
| | $\leq 0.05$ | 0.12 | 63.0 | / | / | / |

Table 1: Transfer learning on Health.

The results are shown in Table 1. First, we observe that some methods are unable to reduce $\Delta_{\mathcal{Z}_0, \mathcal{Z}_1}^{DP}$ below the given threshold. Our method can always reduce the $\Delta_{\mathcal{Z}_0, \mathcal{Z}_1}^{DP}$ sufficiently, but due to our restriction on representations which enables provable upper bounds, we often lose more accuracy than other methods for high $\Delta_{\mathcal{Z}_0, \mathcal{Z}_1}^{DP}$ thresholds. Future work could focus on investigating alternative restricted encoders with better fairness-accuracy tradeoffs in the transfer learning setting.

## 7    CONCLUSION

We introduced FARE, a method for provably fair representation learning. The key idea was that using restricted encoders enables a practical statistical procedure for computing a provable upper bound on unfairness of downstream classifiers trained on these representations. We instantiated this idea with a tree-based encoder, and experimentally demonstrated that FARE can for the first time obtain tight fairness bounds on several datasets, while simultaneously producing empirical fairness-accuracy tradeoffs similar to prior work which offers no guarantees.

REPRODUCIBILITY STATEMENT

To foster reproducibility, all of our code, datasets and scripts are provided in the Openreview discussion. All of our experiments presented in Section 6 can be run using the instructions we provide in the Readme file that accompanies our code. The hyperparameters used for our runs are described in Appendix B and further detailed in the Readme.

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

| Dataset | Training size | Test size | Base rate ($s$) | Base rate ($y$) |
|---|---|---|---|---|
| ACSIncome-CA | 165 546 | 18 395 | 0.46 | 0.64 |
| ACSIncome-US | 1 429 070 | 158 786 | 0.48 | 0.68 |
| Health | 174 732 | 43 683 | 0.35 | 0.68 |

Table 2: Statistics of evaluated datasets.

## A  MATHEMATICAL TOOLS

We first derive Eq. (2). More details can be found in Madras et al. (2018), and here we provide an overview:

$$
\begin{aligned}
\Delta^{DP}_{\mathcal{Z}_0, \mathcal{Z}_1}(g) &= |\mathbb{E}_{z \sim \mathcal{Z}_0}[g(z)] - \mathbb{E}_{z \sim \mathcal{Z}_1}[g(z)]| \\
&= |\mathbb{E}_{z \sim \mathcal{Z}_0}[-g(z)] + \mathbb{E}_{z \sim \mathcal{Z}_1}[g(z)]| \\
&= |\mathbb{E}_{z \sim \mathcal{Z}_0}[1 - g(z)] + \mathbb{E}_{z \sim \mathcal{Z}_1}[g(z)] - 1| \\
&= |2 BA_{\mathcal{Z}_0, \mathcal{Z}_1}(g) - 1|
\end{aligned}
$$

From this, we can argue that we can drop the absolute value and bound the balanced accuracy of $g$ with the balanced accuracy of $h^*$, finally arriving at Equation 2.

Then, we formally state the Hoeffding's inequality and the Clopper-Pearson binomial confidence intervals, used in our upper-bounding procedure in Section 4.

*Hoeffding's inequality (Hoeffding, 1963)*: Let $X^{(1)}, \dots, X^{(n)}$ be independent random variables such that $P(X^{(j)} \in [a^{(j)}, b^{(j)}]) = 1$. Let $\hat{\mu} = \frac{X^{(1)} + \dots X^{(n)}}{n}$ and $\mu = \mathbb{E}[\hat{\mu}]$. It holds that:

$$
P(\mu - \hat{\mu} \geq t) \leq \exp\left(\frac{-2n^2 t^2}{\sum_{i=1}^{n}(b^{(i)} - a^{(i)})^2}\right).
$$

*Clopper-Pearson binomial proportion confidence intervals (Clopper & Pearson, 1934)*: Assume a binomial distribution with an unknown success probability $\theta$. Given $m$ successes out of $n$ experiments, it holds that:

$$
B(\frac{\alpha}{2}; m, n - m + 1) < \theta < B(1 - \frac{\alpha}{2}; m + 1, n - m) \tag{6}
$$

with confidence at least $1 - \alpha$ over the sampling process, where $B(p; v, w)$ denotes the $p$-th quantile of a beta distribution with parameters $v$ and $w$.

## B  DETAILS OF EXPERIMENTAL EVALUATION

In this section we provide details of our experimental evaluation omitted from the main text.

**Datasets**  As mentioned in Section 6, we perform our experiments on ACSIncome (Ding et al., 2021) and Health (Kaggle, 2012) datasets. In Table 2 we show some general statistics about the datasets: size of the training and test set, base rate for the sensitive attribute $s$ (percentage of the majority group out of the total population), and base rate for the label $y$ (accuracy of the majority class predictor).

ACSIncome is a dataset recently proposed by Ding et al. (2021) as an improved version of UCI Adult, with comprehensive data from US Census collected across all states and several years (we use 2014). The task is to predict whether an individual's income is above \$50,000, and we consider sex as a sensitive attribute. We evaluate our method on two variants of the dataset: ACSIncome-CA, which contains only data from California, and ACSIncome-US, which merges data from all states and is thus significantly larger but also more difficult, due to distribution shift. 10% of the total dataset is used as the test set. We also use the Health dataset (Kaggle, 2012), where the goal is to predict the Charlson Comorbidity Index, and we consider age as a sensitive attribute (binarized by thresholding at 60 years). For this dataset perform the same preprocessing as Balunović et al. (2022), and use 20% of the total dataset as the test set.

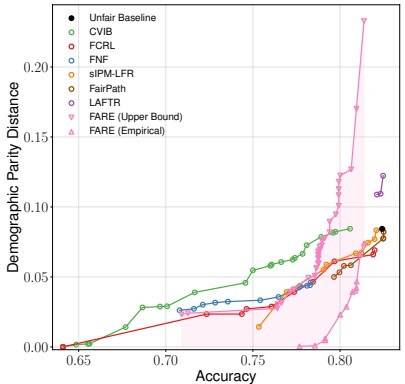 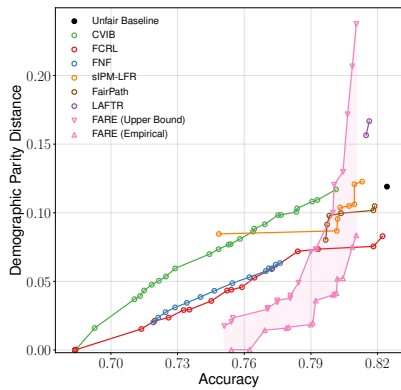

Figure 6: Extended evaluation on ACSIncome-CA (left) and ACSIncome-US (right).

**Evaluation procedure**   For our main experiments, as a downstream classifier we use a 1-hidden-layer neural network with hidden layer size 50, trained until convergence on representations normalized such that their mean is approximately 0 and standard deviation approximately 1. We train the classifier 5 times and in our main figures report the average test set accuracy, and the maximal DP distance obtained, following the procedure of Gupta et al. (2021).

**Hyperparameters**   For baselines, we follow the instructions in respective writeups, as well as Gupta et al. (2021) to densely explore an appropriate parameter range for each value (linearly, or exponentially where appropriate), aiming to obtain different points on the accuracy-fairness curve. For CVIB, we explore $\lambda \in [0.01, 1]$ and $\beta \in [0.001, 0.1]$. For FCRL on ACSIncome we explore $\lambda = \beta \in [0.01, 2]$, and for Health $\lambda \in [0.01, 2]$ and $\beta = 0.5\lambda$. For FNF, we explore $\gamma \in [0, 1]$. For sIPM-LFR, we use $\lambda \in [0.0001, 1.0]$ and $\lambda_F \in [0.0001, 100.0]$, extending the suggested ranges. For FairPath we set the parameter $\kappa \in [0, 100]$. Finally, for LAFTR we use $g \in [0.1, 50]$, extending the range of $[0, 4]$ suggested by (Gupta et al., 2021). We adjust the parameters for transfer learning whenever supported by the method.

For FARE, there are four hyperparameters: $\gamma$ (used for the criterion, where larger $\gamma$ puts more focus on fairness), $\bar{k}$ (upper bound for the number of leaves), $\underline{n_i}$ (lower bound for the number of examples in a leaf), and $v$ (the ratio of the training set to be used as a validation set). Note that all parameters affect accuracy, empirical fairness, and the tightness of the fairness bound. For example, larger $\underline{n_i}$ is likely to improve the bound by making Lemma 2 tighter, as more samples can be used for estimation. For the same reason, increasing $v$ improves the tightness of the bound, but may slightly reduce the accuracy as fewer samples remain in the training set used to train the tree. In our experiments we investigate $\gamma \in [0, 1]$, $\bar{k} \in [2, 200]$, $\underline{n_i} \in [50, 1000]$, $v \in \{0.1, 0.2, 0.3, 0.5\}$. For the upper-bounding procedure, we always set $\epsilon = 0.05$, $\epsilon_b = \epsilon_s = 0.005$, and thus $\epsilon_c = 0.04$. Finally, when sorting categorical features as described in Section 5, we use $q \in \{1, 2, 4\}$ in all cases.

**Omitted details of additional experiments**   For the experiment with alternative classifiers (Fig. 5, left) we explore the following classifiers: (i) 1-hidden-layer neural network (1-NN) with hidden layer sizes 50 and 200, (ii) 2-NN with hidden layers of size $(50, 50)$, as well as $(200, 100)$, (iii) logistic regression, (iv) random forest classifier with 100 and 1000 estimators, (v) decision tree with 100 and an unlimited number of leaf nodes. We train all these classifiers with a standardization preprocessing step as described above. We further train one variant of 1-NN, 2-NN, random forest, and logistic regression, on unnormalized data. All described models are trained both to predict the task label $y$, and to maximize unfairness, i.e., predict $s$, leading to 24 evaluated models.

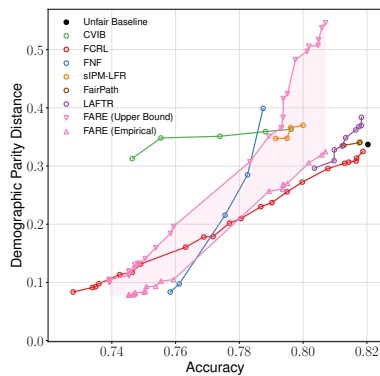

Figure 7: Extended evaluation of FRL methods on Health.

| $y$ | $\Delta^{DP}_{\mathcal{Z}_0,\mathcal{Z}_1}$ | $T$ | FARE | FCRL | FNF | sIPM |
|---|---|---|---|---|---|---|
| MIS | $\leq 0.30$ | 0.64 | 79.3 | 78.6 | 79.2 | 79.8 |
| | $\leq 0.20$ | 0.64 | 79.3 | 78.6 | 78.9 | 79.8 |
| | $\leq 0.15$ | 0.64 | 79.3 | 78.6 | 78.9 | 79.6 |
| | $\leq 0.10$ | 0.48 | 78.8 | 78.6 | 78.9 | 79.0 |
| | $\leq 0.05$ | 0.54 | 78.7 | 78.6 | 78.7 | 78.6 |
| NEU | $\leq 0.30$ | 0.64 | 73.2 | 72.4 | 71.9 | 78.8 |
| | $\leq 0.20$ | 0.64 | 73.2 | 72.4 | 71.9 | 76.6 |
| | $\leq 0.15$ | 0.64 | 73.2 | 72.4 | 71.8 | 73.2 |
| | $\leq 0.10$ | 0.64 | 73.2 | 72.2 | 71.8 | / |
| | $\leq 0.05$ | 0.42 | 72.1 | 71.4 | 71.7 | / |
| ART | $\leq 0.30$ | 0.41 | 74.4 | 70.7 | 68.9 | 78.3 |
| | $\leq 0.20$ | 0.41 | 74.4 | 70.7 | 68.9 | 78.3 |
| | $\leq 0.15$ | 0.46 | 74.2 | 70.1 | 68.9 | / |
| | $\leq 0.10$ | 0.23 | 69.5 | 69.6 | 68.7 | / |
| | $\leq 0.05$ | 0.23 | 69.5 | 69.5 | 68.5 | / |
| MET | $\leq 0.30$ | 0.47 | 74.0 | 72.5 | 76.2 | / |
| | $\leq 0.20$ | 0.46 | 69.8 | 69.2 | 75.0 | / |
| | $\leq 0.15$ | 0.33 | 68.7 | 67.9 | 73.2 | / |
| | $\leq 0.10$ | 0.12 | 66.1 | 66.7 | 73.2 | / |
| | $\leq 0.05$ | 0.12 | 66.1 | 65.3 | / | / |
| MSC | $\leq 0.30$ | 0.56 | 71.3 | 70.5 | 73.5 | 77.6 |
| | $\leq 0.20$ | 0.53 | 67.2 | 70.5 | 73.0 | / |
| | $\leq 0.15$ | 0.12 | 63.0 | 69.7 | / | / |
| | $\leq 0.10$ | 0.12 | 63.0 | 69.0 | / | / |
| | $\leq 0.05$ | 0.12 | 63.0 | / | / | / |

Table 3: Extended results of transfer learning experiments on Health.

For transfer learning (Table 1), the five transfer tasks represent prediction of the following attributes from the Health dataset: MISCHRT (MIS), NEUMENT (NEU), ARTHSPIN (ART), METAB3 (MET), MSC2A3 (MSC).

## C  EXTENDED RESULTS

We provide the extended results of our main experiments, including two originally excluded methods, LAFTR and FairPath in Fig. 6 and Fig. 7, corresponding to Fig. 3 and Fig. 4.

Additionally, in Table 3 we provide extended results of our transfer learning experiments, showing the accuracy values for thresholds $\Delta^{DP}_{\mathcal{Z}_0,\mathcal{Z}_1} \in \{0.30, 0.20, 0.15, 0.10, 0.05\}$. We can observe similar trends as shown in Table 1 in the main paper.

## D  ADDITIONAL EXPERIMENTS

In this section, we provide additional experimental results showing the effects which imbalance in sensitive attribute, different downstream classifiers, and dataset size have on FARE.

### D.1  EFFECT OF SENSITIVE ATTRIBUTE IMBALANCE

In this section, we demonstrate empirically what effect an imbalance in the sensitive attribute has on the resulting fairness and accuracy of our proposed method. Let $c$ denote the level of imbalance of each training set (i.e., the number of data points in the larger of the two sensitive classes divided by the total number of data points in the set). For each value of $c$ we are interested in, we sample a random subset of size 49 053 from the original ACSIncome-CA training dataset (out of 165 546 data points in total) and ensuring that the level of imbalance is exactly $c$. We use 49 053 samples, as this is the largest number for which we can have same dataset size for each $c$, thus ensuring the fair comparison.

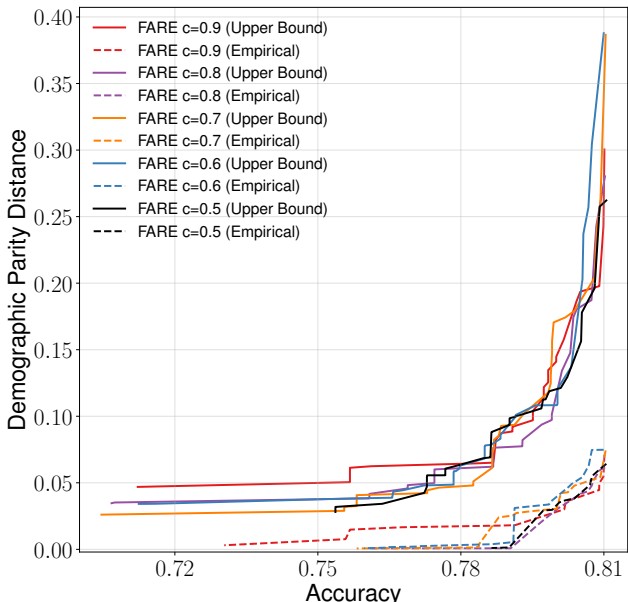

Figure 8: Evaluation of FARE at different levels of imbalance in the sensitive attribute (denoted by $c$) on randomly sampled subsets of ACSIncome-CA of the same size.

We train FARE on each subset separately and show Pareto plots, similar to those in Fig. 3, in Fig. 8. We observe that FARE is very robust to the level of imbalance, as even for very high levels, such as $c = 0.9$, we only see small difference in our Pareto curves (only at the low-accuracy regime).

## D.2 EXPERIMENTS WITH DIFFERENT DOWNSTREAM CLASSIFIERS

In this section, we compare different FRL methods on ACSIncome-CA, similarly to Fig. 3 (left), in the case when different downstream classifiers are used. In Fig. 9, we show the results on 4 additional downstream classifiers:

- Decision Tree with maximum 2500 leaves
- Random Forest using 100 trees
- Logistic Regression
- Two-layer neural network with 50 neurons per layer

We observe that the general trends observed in Fig. 3 (left) for the different FRL methods hold regardless of the downstream classifier choice. We also see that the gap of our method to the maximum achievable accuracy is the smallest when the downstream classifier is a tree. This is unsurprising given that FARE's own representations are based on trees. Further, we see that the more complex feature extraction of FCRL and sIPM-LFR allow them to gain better accuracy over the unfair baseline when using a more simple classifier such as a decision tree.

## D.3 PERFORMANCE GAP ON LARGER DATASETS

In this section, we explore whether the small performance gap we observed in Fig. 3 between FARE's most accurate model and the unfair baseline widens for larger datasets. To this end, we merge two ACSIncome-US datasets from two consecutive years (2014 and 2015) and compare the results to the single year dataset from 2014, shown in Fig. 3 (right). We note that the merged dataset has roughly 2x the number of data points. The comparison between the merged and single-year datasets is shown in Fig. 10. We observe almost no difference between the results on the two datasets for the unfair baseline as well as the empirical and provable fairness of our method. This suggests that the complexity of the dataset is a more important factor than the data volume for the observed performance gap.

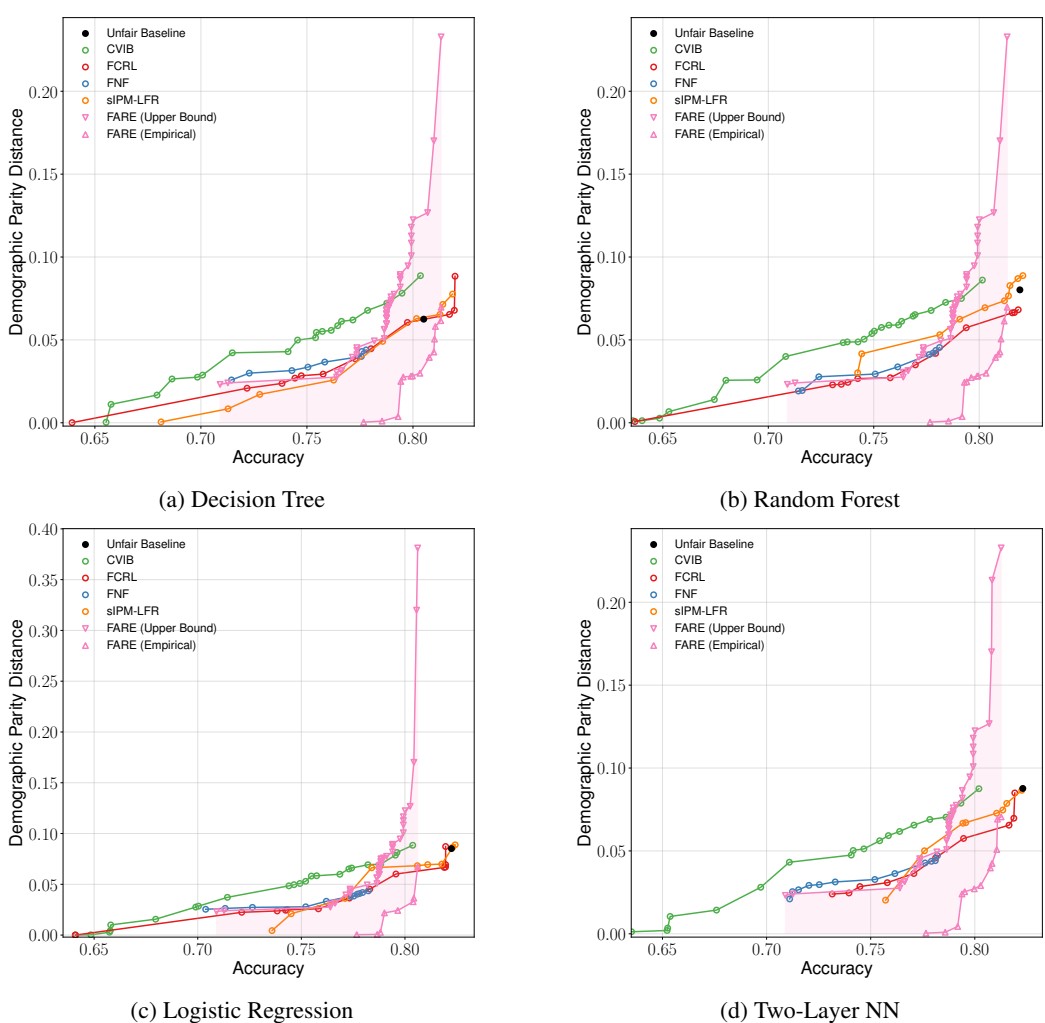

Figure 9: Comparison between different downstream classifiers on different FRL methods on ACSIncome-CA.

# E DIFFERENCES BETWEEN THEORETICALLY-PRINCIPLED FRL AND PROVABLY FAIR REPRESENTATION LEARNING

In this section we discuss the differences between theoretically-principled fair representation learning, which is most of the prior work, and provably fair representation learning, which is our method.

**Provably fair representation learning** Here, we first restate what kind of provable guarantee we want to obtain for our learned representations.

*We use a finite set of $n$ datapoints $\{(\boldsymbol{x}^{(j)}, s^{(j)})\}$ to learn representations $\boldsymbol{z}^{(j)} = f(\boldsymbol{x}^{(j)}, s^{(j)})$ such that the DP distance of any classifier $g$ trained on these representations is bounded by some value $T$ with confidence $1 - \epsilon$. We consider practical scenarios, requiring 95% confidence, and $n$ ranging from 100 000 to roughly 1 000 000.*

Note that this is the only type of guarantee that can provide an assurance to a practitioner that their representations are indeed fair. The practitioner can only benefit from a provable fairness guarantee that is obtained using a finite number of samples, and bounds that are obtained under stronger assumptions such as perfect convergence of the training procedure are less useful.

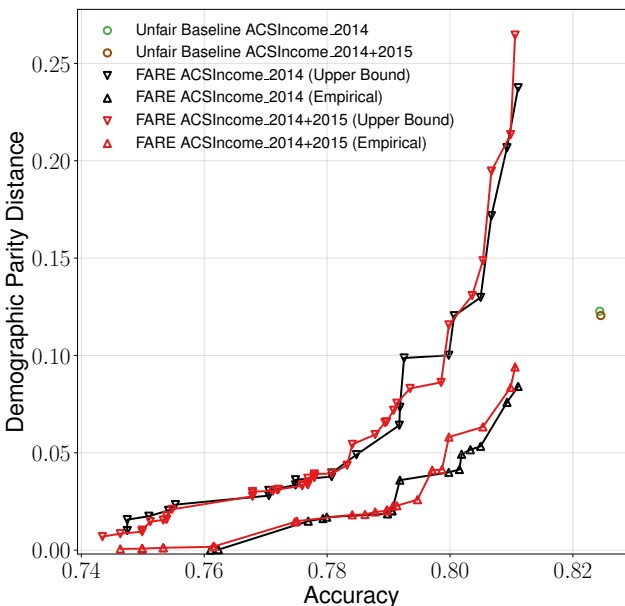

Figure 10: Comparison of the performance gap between FARE and the Unfair Baseline on ACSIncome-US for a single year and two years.

**Theoretically-principled FRL**    We now discuss theoretically-principled FRL methods from prior work and, as argued in Section 2, explain why they cannot achieve the guarantee of provable fairness using a finite number of samples, described above. These are the methods which use sound theory as a basis for their representation learning algorithms. There are several works that compute bounds on DP distance using total variation, mutual information, and other techniques. However, to make the training practical they replace these bounds with approximations that cannot yield high-probability certificates with a finite number of samples, which is what we call provably fair representation learning above.

For example, Madras et al. (2018) bound demographic parity using the accuracy of the optimal adversary for predicting a sensitive attribute from the learned representations (which is also closely connected to the TV distance between the representations). However, even in the case of known input distribution it is not possible to compute the optimal adversary for the learned representations when using standard feedforward neural network encoders (this is discussed in more detail in Balunović et al. (2022)). In this direction, Feng et al. (2019) bound the accuracy of the optimal Lipschitz continuous adversary (though the general optimal adversary does not have to be Lipschitz continuous) using Wasserstein distance, which again cannot be provably bounded for finite number of samples. Kairouz et al. (2022) formulate the optimal adversary for each type of the reconstruction loss, but then explain that it is not possible to compute those in practice (except for the restricted case when input distribution is e.g., a mixture of Gaussians), and they simply use it as a motivation for min-max optimization in training. Perhaps the most advanced work in this direction is Balunović et al. (2022) which proposes a new architecture based on normalizing flows that essentially allows pushing the input distribution through the encoder to obtain the distribution of the latent representations which can then be used to estimate TV distance. However, in practice the input distribution is unknown and their certificate only holds for the estimated input distribution. Zhao et al. (2020) also show that the TV distance can bound the DP distance, but they provide no finite sample certificate. They rely on solving the min-max optimization problem where the objective is approximated and then approximately optimized using SGD. Thus, it is impossible to know whether the resulting representations are indeed fair for any downstream classifier.

Another idea is based on the fact that the DP distance can be bounded with an expression involving mutual information between the representations and sensitive attribute. For example, Gupta et al. (2021) provide one such bound, but they use it as a motivation for their training algorithm and do not provide a way to compute this bound on finite sample dataset. Kim et al. (2022) bound the

DP distance using IPM (integral probability metric), which again cannot be computed with high probability using finite sample dataset. Tan et al. (2020) use classical sufficient dimension reduction framework to construct representations as subspaces of reproducing kernel Hilbert space (RKHS), and apply this to obtain fair Gaussian processes, but they assume existence of a fair subspace which might not generally hold. Similarly, Grünewälder & Khaleghi (2021) generate features oblivious to sensitive attributes using Hilbert-space-valued conditional expectation and relaxed optimization of MMD criterion, which cannot provably guarantee fairness. All of these works fall into the category of theoretically-principled FRL, and they cannot provide a high-probability certificate of fairness using a finite number of samples.

