# OpenReview forum: "FARE: Provably Fair Representation Learning"
_ICLR.cc/2023/Conference — Submitted to ICLR 2023_

### Official Review · Reviewer_bbjC · 2022-10-24

**Confidence:** 3
**Correctness:** 3
**Technical Novelty And Significance:** 2
**Empirical Novelty And Significance:** 2
**Recommendation:** 3

**Clarity, Quality, Novelty And Reproducibility:**

Quality: the theoretical results rely on the choice of the encoder. Theoretical soundness can be improved by explicitly stating those implicit assumptions.

Novelty: comparison with the related work mentioned above is needed to justify the novelty.


**Strength And Weaknesses:**

[Strengthes]
1. The proposed restricted encoder is conceptually flexible and could potentially generalize to a large family of models.
2. The objective based on optimal adversary could be useful for several other fair learning problems.
3. Empirical results show that the proposed method attains improved fairness-accuracy tradeoffs than several popular approaches.

[Weaknesses]
- Novelty: the authors claim to provide the first provable FRL upper bound; however, provable FRL has been widely studied in literature:

[1] Donini et al.,  Empirical risk minimization under fairness constraints, NeurIPS 2018

[2] Tan et al., Learning Fair Representations for Kernel Models, AISTATS 2020

[3] Grünewälder et al., Oblivious Data for Fairness with Kernels, JMLR 2021.

The problem setup of the paper is very similar to those considered in the above papers. Thus, novelty of the paper may not be justified without a comparison with the above related work.

- Soundness: the proposed approach makes implicit  i.i.d. assumption for the data and sensitive attributes, and the results rely on the specific choice of the encoder. While the approach conceptually makes sense, the function class of the encoder is typically unknown for deep architectures. Thus, the results, which rely on a perfect encoder, may not hold in general.


**Summary Of The Paper:**

This paper proposes a tree-based encoder for fair representation learning (FRL). The encoder transforms the input data such that downstream classifiers using the transformed data achieve some fairness guarantees. The authors focus on demographic parity as the fairness criterion in binary classification settings.




**Summary Of The Review:**

I have some doubts regarding the novelty and soundness of the proposed approach (see the comments above).

Questions:
- The related work [1] provides provable guarantees on the demographic parity. How does the result presented in the paper differ from those?
- The restricted encoder is similar to the SDR approach (or sliced inverse regression) described in [2]. What are the advantages of the tree encoder?

---

> ### Author Response · Authors · 2022-11-13
> **Response**
>
> We thank the reviewer for their valuable comments. We provide our responses below.
>
> **Q1: Are the novelty claims in the paper justified, given the works of Donini et al. (2018), Tan et al. (2020), Grünewälder et al. (2021)?**
>
> Yes, we believe our novelty claims are justified. The work of Donini et al. (2018) does not fall into the category of fair representation learning as it is an in-processing method, similarly as Feldman et al. (2015) and Celis et al. (2019) mentioned in our related work. The works of Tan et al. (2020) and Grünewälder et al. (2021) are FRL methods based on kernels that fall into the category of theoretically-principled FRL (see Q1 in our response to reviewer *2duo*). These methods use theory as a basis for the representation learning algorithms, but they cannot obtain high-probability certificates that can be computed using a finite number of samples, which is what we call provably fair representation learning. Note that this is the only type of guarantee that can provide an assurance to a practitioner that their representations are indeed fair.
>
> To clarify this in more detail, we have introduced Appendix E which elaborates on the difference between theoretically-principled FRL and provably fair representation learning, and discusses each theoretically-principled FRL method in more detail.
>
> **Q2: The function class of the encoder is typically unknown for deep architectures. Do you rely on a perfect encoder?**
>
> No, we do not assume the existence of some perfect encoder. Moreover, we are not proving fairness for existing, unknown encoders (that could be DNNs). Instead, we are in a practical setup where the same party (i) chooses the encoder class and training algorithm, (ii) trains the encoder, (iii) attempts to analyze the fairness properties of the resulting representations. In this setting, we propose a new architecture for which there is a way to prove the fairness of resulting representations and we provide a training algorithm (Section 5) to train an encoder that balances fairness and accuracy.
>
> **Q3: How does the work of Donini et al. (2018) differ from the results in your paper?**
>
> As explained in Q1 above, the work of Donini et al. (2018) does not fall into the category of fair representation learning as it is an in-processing method, similarly as Feldman et al. (2015), Celis et al. (2019) mentioned in our related work.
>
> **Q4: What are the advantages of the decision tree compared to SDR (Tan et al. (2020))?**
>
> SDR falls into the general category of theoretically-principled FRL (see above), but does not enable provably fair representation learning, as discussed in more detail in Appendix E. Provably fair representation learning can so far be achieved only using our method explained in Section 4. Our method is compatible with any kind of a restricted encoder (an encoder which maps inputs into a finite number of representations). SDR does not result in a restricted encoder as it maps representations into a continuous subspace. As noted in our response to reviewer *2duo* (Q5), the main advantage of the decision tree in particular, is that it allows us to control key parameters of the proof in Section 4: the number of leaves and the number of samples per leaf. When adequately controlled, these parameters enable tight fairness certificates for a finite number of samples, as shown in our experimental results in Section 5.
>
> **Q5: Should you go beyond the i.i.d. assumption for the data and sensitive attributes?**
>
> No, we believe tackling the provably fair representation learning problem under the i.i.d. assumption is a strong contribution. We follow the standard setup in the research area of fair representation learning (e.g. Madras et al. (2018) and all other works in the area), and even broadly in most of machine learning, and assume that the data is i.i.d. Future work can explore interactions of FARE with many other concerns such as other fairness notions, multiple sensitive attributes, and also non-i.i.d. data, i.e., distribution shifts—while interesting, these concerns are not fundamental for the line of work of provable fairness.

---

> > ### Comment · Reviewer_bbjC · 2022-11-23
> > **After rebuttal**
> >
> > Thanks for addressing my comments. Both the works [1-2] provide finite-sample fairness bounds. Fairness-accuracy tradeoffs are also studied in these related work. Thus, the statement "Provably fair representation learning can so far be achieved only using our method explained in Section 4." do not make sense to me. As also pointed out by reviewer 2duo, the theoretical contribution of the paper is incremental, especially for justifying a provably fair method. For these reasons, my score is unchanged.

---

> > > ### Author Response · Authors · 2022-11-24
> > > **Cited works do not affect our contribution**
> > >
> > > The reviewer has simply restated the claim that two cited works imply that the contribution of FARE is incremental. As previously detailed, this claim is false. The key contribution of FARE, the first provable FRL method, holds, and is not affected by two cited works.
> > >
> > > To reiterate our prior arguments regarding the two cited works:
> > >
> > > - Donini et al. (2018) is an in-processing method, considering a fundamentally different setting from ours (pre-processing/FRL). The differences between pre-, in-, and post- processing settings and key advantages/use-cases of FRL are universally acknowledged in the field and discussed in many works (e.g. see page 63 in the book [1] for a brief overview). As these research directions are largely independent, as usual in this field, our contribution holds despite Donini et al. (2018).
> > >
> > > - Tan et al. (2020) belongs to theoretically-principled FRL methods, as explained above and in our Appendix E. While this and similar methods undoubtedly provide more value than purely empirical FRL, they do not enable a practitioner in a realistic setting to obtain a practical high-probability fairness certificate using a finite dataset. FARE fundamentally differs as it enables this for the first time, thus our contribution is significant.
> > >
> > > We encourage the reviewer to engage with our arguments and we hope to resolve this misunderstanding. To enable a productive discussion the reviewer should:
> > >
> > > - Elaborate on the highly non-standard claim that the in-processing method of Donini et al. (2018) affects the contributions of FARE.
> > > - Describe more concretely, pointing to relevant parts of the paper, how can a practitioner with a finite dataset apply Tan et al. (2020) to obtain a high-probability population-wide fairness certificate, as well as point out evidence that bounds obtained this way attain practical values on real-world datasets, similar to what we demonstrate in our Figures 3 and 4.
> > >
> > > [1] Barocas, Solon, Moritz Hardt, and Arvind Narayanan. "Fairness in machine learning: Limitations and Opportunities"

---

> > > ### Author Response · Authors · 2022-12-06
> > > **Reminder**
> > >
> > > As the discussion phase is ending soon, we encourage the reviewer to respond to our last comment, and address the two discussion points that are open.

---

### Official Review · Reviewer_8jvQ · 2022-10-30

**Confidence:** 2
**Clarity, Quality, Novelty And Reproducibility:** The authors' contribution is clear an…
**Correctness:** 4
**Technical Novelty And Significance:** 3
**Empirical Novelty And Significance:** 4
**Recommendation:** 8

**Strength And Weaknesses:**

This paper provides an excellent overview of the prior work on FRL, highlighting their strength and weakness.  The FARE procedure is practical and is backed by theoretical guarantees. The numerical experiments confirm its superiority over the competitors.

**Summary Of The Paper:**

Fair representation learning (FRL) is a popular class of methods aiming to produce
fair classifiers via data preprocessing, but accuracy-fairness tradeoffs are challenging to achieve using the current toolbox.  To this end, the authors develop

- a practical statistical procedure that, for restricted encoders, upper bounds the unfairness of any downstream classifier trained on their representations.
- an end-to-end FRL method, FARE, that instantiates this approach with a fair decision tree
encoder and applies it to augment the representations with a tight provable upper bound on the unfairness of any downstream classifier.

The theoretical results are validated using many comparative experiments.

**Summary Of The Review:**

The authors have developed a practical procedure for FRL, called FARE, with strong theoretical guarantees.  The numerical experiments confirm its promise relative to the alternatives.

---

> ### Author Response · Authors · 2022-11-13
> **Response**
>
> We thank the reviewer for positive comments on our work and a favorable recommendation; we are happy to answer any potential follow-up questions.

---

### Official Review · Reviewer_qvc4 · 2022-10-31

**Confidence:** 2
**Clarity, Quality, Novelty And Reproducibility:** 1. They claim to be the first to prov…
**Correctness:** 4
**Technical Novelty And Significance:** 3
**Empirical Novelty And Significance:** 3
**Recommendation:** 5

**Strength And Weaknesses:**

1. Providing provably fairness is significant and useful.
2. The evaluation part is sound.


**Summary Of The Paper:**

1. The motivation is to tackle the accuracy-fairness tradeoffs of fair representation learning (FRL), indicating the need of providing provable upper bounds on unfairness of downstream classifiers.
2. They propose Fairness with Restricted Encoders (FARE), the first FRL method with provable fairness guarantees. They restrict the representation space ( i.e., limiting possible representations to a finite set {z 1 , . . . , z k } ) of the (upstream) decision-tree-based encoder.
3. Experiment results on various datasets demonstrate FARE can produce tight upper bounds.


**Summary Of The Review:**

Updated response after the second round discussion: the concern is the likelihood of missing literature of provable FRL and evaluation against such work. I am convinced after the discussion that the paper could be improved with the revision addressing this concern.

---

> ### Author Response · Authors · 2022-11-13
> **Response**
>
> We thank the reviewer for their comments and for raising several interesting questions. We provide our responses below.
>
> **Q1: How tight is the FARE’s provable upper bound on unfairness compared with the maximum unfairness some downstream classifier can exhibit when trained on FARE’s representations.**
>
> The experiment in Figure 5 (middle) explores the tightness of the provable upper bound. To compute the maximum possible unfairness one can note that a model can classify each of the $k$ possible representations to either 0 or 1, which results in $2^k$ possible classifiers (note that this is another benefit of our method and does not work for other representation learning methods). There we can observe that the gap between the upper bound and the maximum DP distance is roughly 0.01, demonstrating that our bound is quite tight, even when estimated from a relatively small number of samples.
>
> If our interpretation of the reviewer’s question does not match what the reviewer had in mind, we are happy to discuss this issue further.
>
> **Q2: Could you explain how equation (2) is obtained?**
>
> The proof of Equation 2 can be found in Section 5.1 of Madras et al. (2018). We have now also provided proof sketch in our Appendix A.
>
> **Q3: For the fairness-aware categorical splits, is rotating the order of categories affecting the split results hence the fairness?**
>
> No, the order of the categories does not affect the split results and fairness.

---

> ### Author Response · Authors · 2022-12-07
> **Actionable feedback would be appreciated**
>
> We have noticed that the reviewer has updated their grade from 8 to 5, requiring evaluation against other provable FRL work. Note that we have already extended the paper to reference all works suggested by other reviewers in our Sec. 2 and added an extensive discussion in App. E. However, as we are not aware of any prior provable FRL work that we can directly evaluate against, we are unable to translate the reviewer's comment into actionable feedback, and we are unsure how to proceed. We would be grateful if the reviewer can provide concrete references, enabling us to improve our manuscript.

---

### Official Review · Reviewer_2duo · 2022-11-03

**Confidence:** 5
**Correctness:** 2
**Technical Novelty And Significance:** 2
**Empirical Novelty And Significance:** 2
**Recommendation:** 3

**Clarity, Quality, Novelty And Reproducibility:**

Overall the paper is clear and easy to follow.


**Strength And Weaknesses:**

Strength:
I didn't check the most recent literature extensively, but the idea of using decision trees as the feature encoders in learning fair representations appears new to me.

Weakness:
Although the application of decision trees as feature encoders in learning fair representations is new, this contribution alone is too incremental to grant acceptance into a major conference like ICLR. Perhaps more importantly, there are quite a few statements in the paper are too strong to be accurate. The authors overclaimed the contributions of this work while missing a lot of closely related works that have already provided fairness guarantees for downstream tasks. Let me elaborate.

-   First of all, apparently, the authors appeared to think that the fairness guarantee of downstream tasks from the learned fair representations could only be obtained using the proposed method, but not from existing encoders with neural networks. This is false. No matter what the encoders are, one can always certify a DP gap for any downstream classifiers based on the learned fair representations. To be more specific, the DP gap for any downstream classifiers is given by the TV distance between Z_0 and Z_1. This is a simple consequence of the data-processing inequality for the TV-distance and the fact that the DP gap is nothing but the TV-distance of the predictions over two groups. See [1] Proposition 3.1 for more details. [1] also discusses the relationship between the TV-distance and the optimal balanced error rate, but is missing from the discussion.

-   In light of the above argument, the real contribution of this work is Lemma 3 which provides a finite sample analysis for discrete distributions using Hoeffding's inequality. However, it still does not justify the "inherent advantages of decision trees as feature encoders". In fact, for any fixed neural network, one can also obtain such high-probability bounds as well by using non-parametric kernel density estimation. The only difficulty here is instead of estimating the probability mass function, which is easier to do because of the finite support, one needs to work with probability density. But in principle it is still doable, and one can then proceed to compute the TV-distance between the estimated density functions from the two groups, which will provide a DP gap for any downstream classifiers over the learned representations.

-   A lot of the important and closely related works in theoretical understanding of learning fair representations are missing. I would suggest the authors to check the Related Work section of [2] for more reference.

More detailed comments:
-   In the abstract, the authors mentioned that "motivated by inherent advantages of decision trees". What are the inherent advantages of decision trees here? Wouldn't any encoder that can provide discrete codes work? For example, consider the approach of Zemel et al.'13, couldn't we just use the deterministic cluster assignment (instead of a probabilistic one) as the codes?

-   In Section 1, "their claim about fairness of the downstream classifiers holds only for the models they considered during the evaluation, and does not guarantee ...". This statement is false. As I explained above, the fairness guarantee on any downstream tasks is guaranteed, and this is just a simple application of the celebrated data-processing inequality.

-   In Section 2, Paragraph "Towards fairness guarantees", "...trained on the representations produced by these methods could have arbitrarily bad fairness". Again, this is simply false. See [1] and [2].

-   Section 4, the discussion of the optimal balanced accuracy is not very deep. At least, one should point out its connection to the TV distance between the two feature distributions (Z_0 & Z_1). See [2] for more details.


[1].    Conditional Learning of Fair Representations
[2].    Fair Representation: Guaranteeing Approximate Multiple Group Fairness for Unknown Tasks


**Summary Of The Paper:**

This paper proposes a method to learn fair representations, where the main focus of fairness is the classic demographic parity condition. Compared with existing methods to achieve this goal, algorithmically, the main difference is that, instead of using rich neural networks as the feature encoders, the authors proposed to use decision trees instead. The features are then encoded as the indices of leaf nodes in the decision tree. Theoretically, the main contribution is to provide a finite sample bound on the probability mass function over the feature space, which is estimated from finite samples.


**Summary Of The Review:**

As mentioned in my detailed comments above, due to the limited technical contribution and the missing discussion with closely related works, I would recommend rejection.

---

> ### Author Response · Authors · 2022-11-13
> **Response (part 1)**
>
> We thank the reviewer for providing thorough feedback. We provide our responses below.
>
> **Q1: Is it possible to certify the DP gap of any downstream classifier by bounding the TV distance between $\mathcal{Z}_0$ and $\mathcal{Z}_1$, regardless of the encoder type?**
>
> No, this is typically not possible. In Appendix E we now more thoroughly describe the difference between *provably fair representation* learning and *theoretically-principled FRL*, which are the methods that generally start from sound theory as a basis for their training algorithms, but make various approximations in order to make the training tractable, making them unable to produce high-probability bounds. Here we provide a summary of the discussion. To begin, we restate what type of certificate we are interested in:
>
> *Given a finite set of $n$ datapoints $\{(x^{(j)}, s^{(j)})\}$, we learn representations $z^{(j)} = f(x^{(j)}, s^{(j)})$ such that the DP distance of any classifier $g$ trained on these representations is provably bounded by some value $T$ with confidence $1 - \epsilon$. We consider practical scenarios, requiring 95\% confidence, with $n$ ranging from 100K to roughly 1M.*
>
> The certificate above is what we refer to as *provably fair representation learning*. Note that only this type of guarantee can provide an assurance to a practitioner, who has a finite dataset, that their representations are indeed fair.
>
> The works that the reviewer mentioned, e.g., Zhao et al. (2019), as well as the works we already discussed in the paper such as Madras et al. (2018), Gupta et al. (2021), Balunovic et al. (2022), Kairouz et al. (2022) are theoretically-principled FRL methods. They use different bounds on DP to motivate their training algorithm, but these cannot be computed with high confidence on a finite sample dataset, and thus they cannot be considered provably fair representation learning.  While the reviewer is correct that the TV of the representations bounds DP, it is generally not possible to obtain high-confidence (non-asymptotic) bounds on TV using a finite number of samples, for an arbitrary encoder. Proposition 3.1 in Zhao et al. (2019), mentioned by the reviewer, merely states that if the TV is 0 we can bound the fairness of a downstream classifier, but as argued above, this does not provide a way to compute a high-confidence bound on the TV using finite data. Similar approach was explored by FNF (Balunovic et al., 2022), and works only under the strong assumption of knowing the input distribution, as discussed in our related work.
>
> To summarize, while prior work has proposed several methods for theoretically-principled FRL, our work is the first one which provides a finite sample guarantee as described above.
>
> We hope this helps clarify the misunderstanding—we are happy to continue the discussion further if the reviewer has follow-up concerns.
>
> **Q2: Can you obtain high probability bounds on DP when using NN encoders by first estimating the density of the latent representations with KDE (kernel density estimation), and then proceed by computing the TV (total variation) between estimated densities of the two groups?**
>
> A: No, this is not possible in the general case. In principle, estimating densities with high probability using KDE requires strong assumptions on the input distribution. For example, there are bounds that apply when the probability density function of the input distribution is Lipschitz [1] or $\alpha$-Holder [2] continuous, and guarantees depend on knowing e.g., the Lipschitz constant. Thus, it is not feasible for a practitioner to compute such guarantees as the only thing they have is a finite dataset.
>
> Besides, even under these strong assumptions, another significant limitation of this approach is that such bounds only work for low-dimensional representations.  For example, in the work of Jiang et al. [2] the error in the estimate depends exponentially on the dimension, meaning that such an estimate would require a large number of samples to achieve any reasonable error, even for small dimensions that are typically used for the representations in prior work (10-50).
>
> [1] Density Estimation, https://www.stat.cmu.edu/~larry/=sml/densityestimation.pdf
>
> [2] Jiang, Heinrich. "Uniform convergence rates for kernel density estimation." International Conference on Machine Learning. PMLR, 2017.

---

> ### Author Response · Authors · 2022-11-13
> **Response (part 2)**
>
> **Q3: Is the statement in Section 1, "their [prior methods’] claim about fairness of the downstream classifiers holds only for the models they considered during the evaluation, and does not guarantee ...", false (see [1, 2])?**
>
> No, our statement is indeed correct and it is a well established fact in the FRL research community. There has been plenty of work (Xu et al., 2020, Song & Shmatikov, 2020, Gupta et al., 2021, Balunovic et al. 2022) demonstrating that it is possible to train classifiers with higher unfairness than the empirical measurements of fair representations suggested by the existing FRL methods. This also holds for theoretically-principled FRL methods (e.g. Madras et al. 2018), and the works [1, 2] suggested by the reviewer also fall into this category and may suffer from the same problem.
>
> For example, the suggested work of Zhao et al. [1] relies on solving the min-max optimization problem in their Equation 2. The objective is approximated in Equation 3.4 and then approximately optimized using SGD. Thus, it is impossible to know whether the resulting representations are indeed fair for any downstream classifier.
>
> **Q4: Did you miss a lot of important related work on FRL?**
>
> No, we do not believe so. We would like to point out that other reviewers mention that we provide an extensive literature survey (*R_1zmn*) and an excellent overview of the prior work on FRL (*R_8jvQ*). FRL is a very large research area and it is not possible to discuss every paper. Having said that, we have included several works from the related work of Shen et al. (2021) that reviewer suggested into our updated related work section (e.g., McNamara et al. 2017, Zhao et al. 2020).
>
> **Q5: What are the inherent advantages of decision trees here? Wouldn't any encoder that can provide discrete codes work? Could you adapt LFR to this setting?**
>
> Our provable fairness bounds, which are our main contribution, are compatible with any restricted encoder (e.g., see beginning of Section 4). We find decision trees the most suitable, as they allow us to control the number of samples in each leaf and the number of leaves itself, which are crucial parameters for the tightness of our bound. Prior work has found that even the original version of LFR does not work reliably (e.g., see Gupta et al. 2021), thus we expect that the less expressive variant (which uses the deterministic cluster assignment instead of a probabilistic one), while in principle simple to implement, would not produce good results. For this reason, we only compared with more recent FRL methods.
>
> **Q6: Could you point out the connection between balanced accuracy and TV distance?**
>
> Yes, we have now cited the work of Shen et al. (2021) and pointed out this connection at the end of the first paragraph in Section 4.

---

> ### Author Response · Authors · 2022-11-24
> **Follow-up**
>
> We thank the reviewer once again for their feedback. We encourage the reviewer to reassess FARE in light of our responses and updates, and reach out with any remaining concerns and questions; we are happy to continue the discussion.

---

> ### Comment · Reviewer_2duo · 2022-11-24
> **Many overclaims and inaccurate statements still hamper the paper**
>
> I thank the authors for the response and the updated manuscript. However, as also pointed out by reviewer bbjC, I still stand with my original review that there are too many overclaims and missing/inaccurate discussion w.r.t. closely related works. So, I insist that this manuscript cannot be accepted in its current form, and I urge the authors to consider tuning down a bit and discussing their contributions in the context of previous works.
>
> In the rebuttal the authors still claim that "No, this is typically not possible." for other encoders to obtain provable fairness guarantees. As I've mentioned (actually provided a short proof sketch on how to achieve this) in my original review, it IS possible to obtain finite sample bounds for the fairness metric for any type of encoders, as long as the VC dimension of the corresponding function class is finite. So from this perspective, it is unclear to me what's the "inherent advantages of decision trees" as claimed by the authors. Theoretically, under the considered setup, there is NO inherent advantages of decision trees -- any encoder just works, and even for NNs. If your main argument is that decision trees could provide discrete codes, then a simple quantization procedure following any continuous encoders just works.
>
> Also, I have to point out that the updated discussion in Appendix E does not seem accurate to me:
>
> > "We consider practical scenarios, requiring 95% confidence, and n ranging from 100K to roughly 1M."
>
> This is not always possible, even for simple encoders like decision trees, as the sample complexity does not only depend on the sample size, but also the complexity of the hypothesis class. Let's say, my hypothesis class consists of decision trees of more than 100 depths, then it's likely even for $n$ ranging from 100K to 1M, it's not sufficient to obtain a 95% confidence bound.
>
> > "Note that this is the only type of guarantee that can provide an assurance to a practitioner that their representations are indeed fair."
>
> Again, this sentence does not make sense. In the literature there are many types of guarantees one can seek to provide to assure practitioners. What has been discussed and studied in this paper is one type of guarantee, i.e., a high-probability bound, but one can also aim to provide an expectation bound. Neither one dominates the other, they are just two different types of bounds that one can obtain depending on the concrete application scenarios.
>
> > "but these cannot be computed with high confidence on a finite sample dataset, and thus they cannot be considered provably fair representation learning."
>
> Unfortunately, this is still false. In fact, all the above bounds could be computed, as long as the corresponding VC dimension of the encoder class is finite. From this perspective, there is no fundamental difference between an NN encoder and a decision tree encoder. In fact, in certain cases, an NN encoder could have even smaller VC dimension than a decision tree encoder, leading to better and tighter bounds for the confidence parameter. For example, consider a low-dimensional linear NN versus a 100-layer decision tree.
>
> Hence, unless the authors can properly discuss their contributions in light of previous works, I cannot recommend acceptance given that there are too many overclaims that could misguide readers who are not familiar with the line of works on FRL.

---

> > ### Author Response · Authors · 2022-11-26
> > **Continuing the discussion; contributions still hold (part 1/2)**
> >
> > We have attempted to restate the criticism from reviewer’s last reply as four discussion points we hope to jointly resolve.
> >
> > **Point 1**, reviewer states: _“It is not always possible to obtain 95% confidence for n roughly in [100k, 1M], as the required encoders might be too complex.”_
> >
> > Crucially, note that we are naturally interested in analyzing **a particular encoder**, and not whole hypothesis classes, as the former corresponds to a practical setting where a single set of representations from a single encoder are to be used in downstream tasks. With this in mind, while arbitrarily complex problems certainly exist and may require arbitrarily complex models, we agree this statement could more explicitly say that we consider common classification problems established in the relevant fairness literature—our aim was to capture this with the phrase “practical scenarios”. Indeed, as we show in Sec. 6, for all such practical scenarios, and for various accuracy levels, FARE can find a restricted encoder that is not “too complex”, meaning that it can achieve a 95%-confident meaningfully tight upper bound on the DP distance of downstream classifiers. For example, in Fig. 3 (right) we can find an encoder (with hyperparameters as in App. B) for which there is a downstream classifier with ~79% accuracy and all downstream classifiers **provably** have at most ~0.07 DP distance, comparable to empirical values of baselines. We will update the writing to make these points clearer.
> >
> > **Point 2**, reviewer states: _“From the perspective of provability, there is no fundamental advantage of decision trees compared to any other restricted encoder (as defined in Sec. 4).”_
> >
> > We certainly agree, and believe that the current writing regarding our key contributions reflects this, thus this is at most a writing concern. In Sec. 4 where we present our main contribution (the statistical procedure) we clearly state it works on any restricted encoder. However, while the theory is sound, finding an instantiation of a restricted encoder that can successfully balance the {accuracy, fairness, proof confidence} tradeoffs is another big challenge. For our other key contribution (Sec. 5 and 6), we find one such instantiation, and demonstrate that extending and tuning decision trees can lead to good results in practice. Other fundamentally different instantiations surely might exist, but finding them is a major challenge. We expect LFR-based solutions (discussed in Q5) and solutions based on NNs with discretization post-processing, as suggested by the reviewer, to not immediately lead to good performance, but we certainly find this interesting and welcome future work exploring this direction. We did initially consider several solutions, and as noted in Q5, simply chose to focus on a tree-based approach due to convenient properties for our particular use-case (explicit control over parameters). We do not attempt to claim that they are fundamentally superior to other restricted encoders, and we certainly agree that some trees are more complex than some NNs. If the reviewer feels the current writing places too much emphasis on trees, we would appreciate concrete suggestions on how to improve this, and are happy to incorporate them.
> >
> > **Point 3**, reviewer states: _“It is possible to obtain the same type of high-probability proof for an arbitrary finite-VC NN encoder. To do this, we can use the previously proposed procedure (discussed in Q1/Q2 above) with the addition of VC-based bounds on TV.”_
> >
> > Applying VC-based bounds, typically used to bound generalization error, to bound TV and fix the procedure discussed in Q1/Q2 might be possible in theory, but we are not aware of any work doing this, especially as an end-to-end system and in the context of FRL. Can the reviewer point us to such works? Even if possible, we expect this to not have much practical value—the key reason is that (as noted in Point 1) we are interested in analyzing a **fixed encoder**, which in turn provides the bounds on DP distance of **any downstream classifier**. In contrast, we believe the reviewer proposes using VC-based bounds to operate on entire hypothesis classes of encoders, which would inflate the bounds by orders of magnitude and is unlikely to lead to meaningfully tight (as discussed in Point 1) DP bounds on classifiers. In summary, we appreciate the discussion and are happy to adjust some claims (e.g., qualify “impossible” with notes on practicality and existence of complete solutions in prior work). However, we believe that this rephrasing would not diminish the contributions of FARE, as the first end-to-end framework which clearly demonstrated meaningfully tight bounds (and no significant accuracy loss) in common practical settings.

---

> > ### Author Response · Authors · 2022-11-26
> > **Continuing the discussion; contributions still hold (part 2/2)**
> >
> > **Point 4**, reviewer states: _“High-probability bound on the DP distance is not more useful than bounding its expectation.”_
> >
> > We respectfully disagree, and believe the impact of FARE fundamentally relies on this. Namely, the key motivation for fairness is to avoid deployment of unfair classifiers, as even a single unfair classifier can cause harm to a large number of individuals. We find the upper bounds on the DP distance are the only bounds that can help prevent this—bounding the expectation implies nothing about the possibility of training discriminatory classifiers on the representations. Namely, with guarantees that hold in expectation you might still be vulnerable to attacks as in (Xu et al., 2020, Song & Shmatikov, 2020, Gupta et al., 2021, Balunovic et al. 2022)---this is w.h.p. not possible with the bound as given by FARE. As we hope to understand why there is a disagreement, could the reviewer provide concrete examples of referenced scenarios where the expectation bound is strictly more useful?
> >
> > **Summary**: While we wholeheartedly agree that overclaims are an important issue and need to be discussed, we believe our key contributions undoubtedly hold, and no fundamentally incorrect claims about the impact of FARE were made. We encourage the reviewer to address the discussion points, and as noted above we are eager to improve the writing in places which could have been perceived as too strongly worded.

---

> > ### Author Response · Authors · 2022-12-06
> > **Reminder**
> >
> > As the discussion phase is ending soon, we encourage the reviewer to address the open discussion points listed in our previous response; we are also happy to incorporate concrete writing changes that the reviewer feels would better calibrate our claims.

---

### Official Review · Reviewer_1zmn · 2022-11-03

**Confidence:** 2
**Correctness:** 4
**Technical Novelty And Significance:** 3
**Empirical Novelty And Significance:** 3
**Recommendation:** 8

**Clarity, Quality, Novelty And Reproducibility:**

Clarity and quality
--------------------------
The paper clearly introduces the problem, provides extensive literature survey related to different aspects of the paper and clearly separates out these references in section 2.
It provides a demonstrative example in Figure 1 which helps the reader quickly understand the main idea.

The empirical and theoretical validation for the proposed technique are very neatly communicated to the reader. The authors motivate the need for provable guarantees and at the same time demonstrate that these do not come with a significant cost in terms of accuracy.

Novelty
---------
The critical contribution of this paper is in the provability of the fairness guarantees for any down-stream classifier operating on the proposed representations.



**Strength And Weaknesses:**

Strengths
-----------
Clear derivations.
Sufficient experimental validation.
Very clear demonstration of the core concepts even for a casual reader.


Weaknesses
---------------
The classifiers used for empirical validation are fairly simple, which might be suitable for the relatively small datasets being used. However using a diverse set of classifiers for empirical validation might be preferable.


**Summary Of The Paper:**

In this paper authors exploit the use of a restricted encoder to derive a provably fair (group fairness) representation which has the ability to upper bound the unfairness of any down stream classifier. They demonstrate this ability through the use of an optimal adversary, i.e., a classifier which tries to predict the sensitive variable given the representation. Such a bound is computable as the restriction in the encoder leads to a finite set of representations.

**Summary Of The Review:**

This paper contributes to the very critical area of fair representation learning. The extensive literature survey and the clear description of the contributions of proposed technique in contrast with existing literature can make it a very good introduction paper for readers delving into this topic.

The experimental validation could be strengthened by focussing on tasks (at least simulated tasks) where the data sizes are larger. This can allow for the development of more powerful downstream classifiers which might help provide a stronger estimate of accuracy of an unfair classifier and thus the loss in accuracy due to the fair representation.

It would be also helpful if the authors discuss the impact of balance in the training data on the granularity of the restricted representations. It is not uncommon in many domains to be presented with datasets which are highly imbalanced w.r.t. sensitive variables. In such cases Fairness-aware categorical splits could possibly lead to uninformative representations for classification purposes. Adding a discussion on balance of training data would be helpful to readers dealing with such datasets.

---

> ### Author Response · Authors · 2022-11-13
> **Response**
>
> We thank the reviewer for suggesting additional insightful experiments. We provide our responses below.
>
> **Q1: Can you include experiments with a more diverse set of downstream classifiers?**
>
>
> Yes, we have updated our paper with a new experiment with 4 additional downstream classifiers on the ACSIncome-CA dataset in Appendix D.2. The results, shown in Figure 9, are very similar to Figure 3 (left) for all of the different classifiers. Please refer to Appendix D.2 for a more detailed discussion.
>
> **Q2: Does FARE lose more accuracy on larger datasets?**
>
> Yes, we have provided an additional experiment in Appendix D.3 in the latest version of our paper where we combined 2 years from the ACSIncome-US dataset to form a larger dataset. We trained both FARE and the Unfair Baseline on this merged dataset and compared the results to the single-year results in Figure 3 (right). We observed that the results for both FARE and the Unfair Baseline are nearly indistinguishable between merged and single-year datasets. This suggests that the size of the data might not play a big factor in the accuracy loss.
>
> Further, we point out that the recently proposed ACSIncome-US (~1.5M training samples) is larger than all datasets previously used in this field, and our work is among the first ones to include it alongside smaller datasets usually used in prior work (e.g. UCI Adult with ~25k training samples or Heritage Health with ~175k training samples). Therefore, our experiments are in line or even exceed the previous work in this field in terms of scale.
>
> **Q3: Can you show the impact of imbalance w.r.t. the sensitive variables on the accuracy and fairness of FARE?**
>
> Yes, in Appendix D.1 we have now provided the experiments with imbalanced data w.r.t. the sensitive attribute on ACSIncome-CA with different levels of imbalance. We see that our method is robust to imbalance and that results for even severe levels of imbalance (0.9) are only slightly worse than the results for the balanced data. Further, the difference occurs only in the low-accuracy regime.

---

### Author Response · Authors · 2022-11-13
**General response**

We thank the reviewers for their positive and insightful feedback. We are pleased that the reviewers found our contribution of enabling provable fairness significant (*R-qvc4*), literature review excellent (*R-8jvQ*) and extensive (*R-1zmn*), and are satisfied with our experimental evaluation (*R-1zmn, R-qvc4, R-8jvq*). We provide a separate response to each reviewer. We also updated the paper with additional experiments in Appendix D, and thorough discussion of the differences between provably fair (our work) and theoretically principled fair representation learning (most of the prior work) in Appendix E.

---

### Decision · Program_Chairs · 2023-01-20

**Decision:**

Reject

**Justification For Why Not Higher Score:**

Overclaiming and framing of the paper in light of existing work; needs major revision

**Justification For Why Not Lower Score:**

N/A

**Metareview: Summary, Strengths And Weaknesses:**

Reviewers were initially split on this paper, but after discussion, it was clear that some of the initial reviews were overly optimistic. Two experts on fair representation learning (RL) were critical of the characterization of this work as "the first provably fair algorithm", and provided several references to support this claim.

For the sake of the authors, I will try to summarize the issue here: This submission is about finite-sample guarantees for a particular algorithm for fair RL. While the authors have (perhaps cleverly) **defined** "provably fair" in this sense, this is not the common interpretation of the word "provable". Other papers about other approaches to fairness have provable guarantees, although they may not be finite sample or about representation learning specifically. It is also unclear if this is the "first", since it is clear that there is existing work with guarantees. I would suggest avoiding this claim altogether given the amount of work in this rapidly growing area. The results are sufficiently interesting without these (over)claims.

Ultimately, I agree that the paper needs another round of major revision before it can be accepted to a top ML conference. At the very least, the framing of this paper needs revision and a detailed comparison against existing work added to the main paper. Adding experiments against existing methods (whether RL-based or not) would also improve the paper significantly.

**Summary Of Ac-Reviewer Meeting:**

Summarized above.